# Expanding the limits of nuclear stability at finite temperature

Ante Ravlić [1] ✉, Esra Yüksel [2], Tamara Nikšić [1] & Nils Paar [1] ✉

Properties of nuclei in hot stellar environments such as supernovae or neutron star mergers are largely unexplored. Since it is poorly understood how many protons and neutrons can be bound together in hot nuclei, we investigate the limits of nuclear existence (drip lines) at finite temperature. Here, we present mapping of nuclear drip lines at temperatures up to around 20 billion kelvins using the relativistic energy density functional theory (REDF), including treatment of thermal scattering of nucleons in the continuum. With extensive computational effort, the drip lines are determined using several REDFs with different underlying interactions, demonstrating considerable alterations of the neutron drip line with temperature increase, especially near the magic numbers. At temperatures $T \lesssim 12$ billion kelvins, the interplay between the properties of nuclear effective interaction, pairing, and temperature effects determines the nuclear binding. At higher temperatures, we find a surprizing result that the total number of bound nuclei increases with temperature due to thermal shell quenching. Our findings provide insight into nuclear landscape for hot nuclei, revealing that the nuclear drip lines should be viewed as limits that change dynamically with temperature.

The atomic nucleus consists of protons and neutrons, held together by the strongest fundamental force in nature. One of the most important challenges in nuclear physics is to determine the properties of nuclei with extreme neutron-to-proton ratios and the location of the nuclear drip lines, representing the maximum number of nucleons that a nucleus can contain before undergoing proton or neutron emission[1,2]. Determining the nuclear properties away from the valley of beta stability and the limits of nuclear existence can reveal the complex nature of nuclei and force that holds nucleons together[3]. Moreover, it has a direct impact on the production of chemical elements in stars[4–6]. The drip lines set the limits for theoretical description of nuclear masses, $\beta$-decay rates, neutron capture rates, neutrino induced reactions, and other properties required in modeling nucleosynthesis and stellar evolution[7,8]. Extreme astrophysical events, such as core-collapse supernovae[8,9] and neutron star mergers[10–12], attain high enough temperatures to influence the nuclear properties. Furthermore, finite-temperature effects contribute in the evolution of the nuclear $r$-process[7,13,14], as well as nuclear heavy-ion reactions and decays from

excited states[15–17]. Complete understanding of all these phenomena requires knowledge of nuclear properties at finite temperature.

Measurements of nuclei far from the valley of stability are rather challenging. In fact, at zero temperature, the neutron drip line is uncovered only up to $Z = 10$[18]. Therefore, nuclear physicists employ theoretical tools to understand the evolution of the nuclide chart with increasing nucleon numbers and to determine the location of the nuclear drip lines, though mainly limited to zero temperature. In this context, the nuclear energy density functionals (EDF) stand out as successful and reliable microscopic theoretical framework with extrapolation abilities. In recent years, significant amount of work has been done to predict the nuclear drip lines at zero temperature using both non-relativistic[1,19–26] and relativistic (covariant)[2,27–29] EDFs together with gross-model predictions[30–32]. Experimental efforts at finite temperature are mostly confined to investigating collisions of heavy ions, and subsequent decays of compound nuclei, faced with many open questions and still far from their application near the drip line[33]. Several theoretical studies have been conducted to investigate the effect of temperature on nuclear

[1]Department of Physics, Faculty of Science, University of Zagreb, Bijenička c. 32, 10000 Zagreb, Croatia. [2]Department of Physics, University of Surrey, Guildford, Surrey GU2 7XH, UK. ✉e-mail: aravlic@phy.hr; npaar@phy.hr

properties, however microscopic ones have been limited to specific regions of the nuclear chart or have used microscopic models that omit important considerations of particle continuum[34–40], while the others are limited to phenomenology[41]. With the inclusion of temperature in the mean-field calculations, nucleons are thermally scattered into the continuum. If the boundary conditions of continuum states are not properly treated, one finds that the results of the calculations are dependent upon the box size used to discretize the problem[42–44]. This is of special relevance for the weakly-bound nuclei near the nucleon drip lines.

In this work, we address the following questions: (i) How do nuclear properties evolve with increasing temperature across the nuclide chart? and (ii) How does the location of the drip line change as the temperature of the system increases? To resolve these questions, here we introduce a sophisticated microscopic theory of hot nuclei based on a relativistic nuclear energy density functional, including the nuclear deformation, pairing correlation effects, and treatment of nucleon states in the continuum. For the latter, we employ the Bonche-Levit-Vautherin (BLV) continuum subtraction procedure[42,43], that enables us to isolate the bound nuclear states from the continuum (vapor) states, which is essential for a realistic description of nuclei at finite temperature, especially near the drip lines. We present our results using seven relativistic EDFs that allow us to estimate theoretical uncertainties and sensitivity of the drip lines on the symmetry energy of the nuclear equation of state. The analysis of potential energy surfaces provides insight into the temperature evolution of the nuclear shapes, illustrating how nuclei eventually become spherical with temperature increase. To quantify the impact of continuum states we calculate neutron emission lifetimes for hot nuclei and demonstrate their rapid decrease with increasing neutron number and temperature. By mapping the neutron and proton drip lines for nuclei in the temperature range of $T = 0$–$2$ MeV, we show that temperature effects significantly alter their location, especially around shell closures at $N = 82$, $126$, and $184$. Surprisingly, our analysis reveals that at higher temperatures the nuclear landscape is expanding, i.e., at $T \gtrsim 1$ MeV the total number of bound nuclei starts to increase because of the thermal quenching of the shell effects.

## Results
### Two-nucleon drip lines at finite temperature
In its simplest definition, the limit of the nuclear landscape can be determined from the one or two nucleon separation energies ($S_{n(p)}$, $S_{2n(2p)}$). When considering even-even nuclei, the two-neutron drip line is reached when the two-neutron separation energy changes sign, $S_{2n} = E(Z,N) - E(Z, N-2) \geq 0$, with analogous definition for two-proton drip line $S_{2p} = E(Z,N) - E(Z-2, N) \geq 0$, where $E(Z,N) < 0$ is the total binding energy of a nucleus with $Z$ protons and $N$ neutrons. The equivalent definition for the two-neutron(proton) drip line is the condition on neutron(proton) chemical potential (Fermi level). For bound even-even nuclei, the chemical potential is negative, and the drip line is obtained when it changes its sign and becomes positive[22,45].

A straightforward generalization of the two-nucleon drip line definition to finite temperature is achieved by replacing the total binding energy in the expression for $S_{2n(2p)}$ with the free energy $F(Z,N)$. Within the BLV method, we consider the subtracted free-energy $\bar{F}$, which is devoid of the nucleon vapor contribution, and defined as $\bar{F}(Z,N) = \bar{E}(Z,N) - T\bar{S}(Z,N)$, where $\bar{E}$ and $\bar{S}$ are the subtracted total binding energy and entropy. Therefore, at finite temperature, the two-neutron and two-proton drip lines are defined with $S_{2n} = \bar{F}(Z,N) - \bar{F}(Z, N-2) \geq 0$ and $S_{2p} = \bar{F}(Z,N) - \bar{F}(Z-2, N) \geq 0$, respectively. Furthermore, we show that the drip lines at the finite temperature within the BLV prescription, defined by the free-energy difference ($S_{2n(2p)} \geq 0$) and the chemical potential ($\lambda_{n(p)} \geq 0$) are equivalent (Supplementary Fig. 7), confirming the consistency within our model.

The two-neutron(proton) drip lines are shown in Fig. 1 for temperatures $T = 0$, $0.5$, $1.0$ and $2.0$ MeV, corresponding to the temperature range up to $2.3 \times 10^{10}$ kelvin. A finer temperature mesh also containing $T = 0.8$, $1.2$, $1.5$, and $1.8$ MeV is displayed in Supplementary Fig. 10. We show the mean results calculated by using three state-of-the-art relativistic EDFs: DD-ME2[46], DD-PC1[47], and DD-PCX[48], together with the shaded bands representing the corresponding systematic uncertainties. First, we focus on the mean results, denoted by thick colored lines. One immediately observes that the temperature strongly alters the predicted drip lines, especially the ones around shell closures at $N = 82$, $N = 126$, and $N = 184$. As expected, the two-neutron drip line is more influenced by the finite temperature effects compared to the two-proton drip line. In the vicinity of the neutron drip line, nucleons populate higher-energy levels and are more easily coupled with the states in the particle continuum. On the other hand, such coupling is reduced for the proton states due to the increasing height of the Coulomb barrier with increasing proton number. Already at $T = 0.5$ MeV, temperature has a slight effect on drip lines, mainly near the neutron shell closures around $N = 82$, $N = 126$ and $N = 184$.

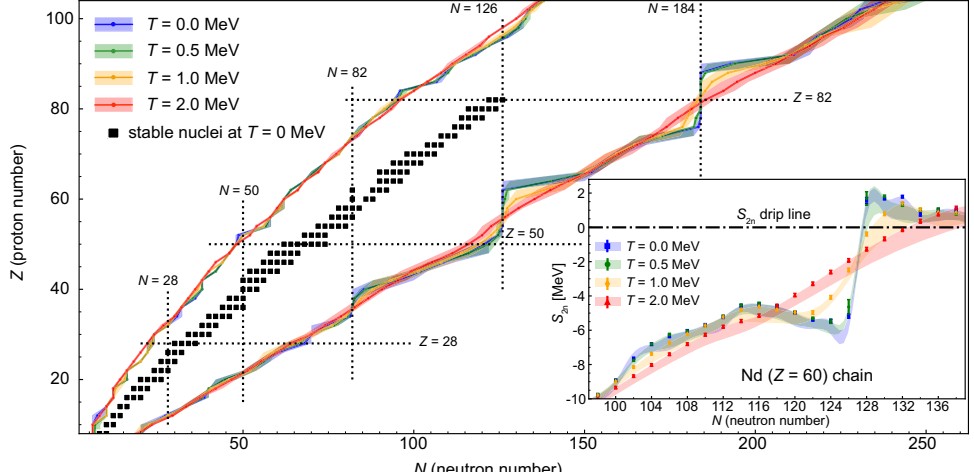

**Fig. 1 | Two-nucleon drip lines at finite temperature.** The evolution of two-neutron and two-proton drip lines, with the corresponding systematic uncertainty bands, for even-even nuclei between $8 \leq Z \leq 104$ at $T = 0$, $0.5$, $1.0$ and $2.0$ MeV. Black squares denote the experimental data for even-even stable nuclei at zero temperature[72]. Proton and neutron shell closure numbers are labeled with vertical and horizontal dotted lines. The inset shows two-neutron separation energy $S_{2n}$ for neodymium isotopic chain ($Z = 60$) at finite temperatures. Colored bands correspond to systematic uncertainties, while the statistical uncertainties for the DD-PCX functional are displayed with the error bars. Black dot-dashed line within the inset denotes the drip line condition $S_{2n} = 0$.

As the temperature increases to $T = 1.0$ MeV, changes in drip lines become more pronounced. At such a high temperature, the pairing correlations vanish for most nuclei, and a transition occurs from the superfluid to the normal states. Furthermore, nuclear deformation tends to decrease with temperature, leading to a transition from the deformed to the spherical state. However, at $T = 1.0$ MeV, we find that a significant number of nuclei still exhibit deformation effects. Shell effects at $N = 82$, $N = 126$, and $N = 184$ are also quenched with additional energy in the environment, and nucleons can bridge the gaps between the neutron shells. At $T = 2.0$ MeV, shell effects are completely washed out, nuclei are mostly spherical, and the neutron drip line is described by a simple straight line. This can be understood on the basis of the Fermi gas model of the nucleus, which predicts a simple temperature dependence of the free energy, $F \sim T^2$[44]. It is also seen that, at high temperatures, the number of bound nuclei decreases before the shell closures, while more nuclei are found to be bound just after the shell closures. The net effect is that the overall number of bound nuclei increases for $T \gtrsim 1$ MeV (Supplementary Table 2).

Different functionals employ different parameterizations of the mean-field Lagrangian together with a different set of observables used to optimize their parameters. Such a variety in the optimization of EDFs leads to systematic errors. It is important to verify that systematic uncertainty allows the finite temperature effect on the drip line to be distinguished. Observing the shaded regions in Fig. 1, we notice that systematic uncertainties on the two-proton drip line are negligible. On the neutron-rich side, the systematic uncertainties are more pronounced. However, the finite temperature effect on the drip line is large enough to allow for a clear distinction between its effect and systematic errors.

To investigate the details behind the finite temperature effects on the two-neutron drip line, in the inset of Fig. 1, we show the two-neutron separation energy $S_{2n}$ for even-even isotopes of neodymium ($Z = 60$). We present results with shaded systematic uncertainty bands originating from the variance between different functionals. On top of this, for one particular functional (DD-PCX[48]), we also show the statistical errors (colored error bars) originating from the uncertainties in the EDF parameters. One observes that statistical uncertainties are much lower in comparison to systematic ones and can be neglected. At $T = 0$ MeV, $^{186}$Nd is the drip-line nucleus. Such a result is in accordance with $N = 126$ being the closed shell number. At $T = 0.5$ MeV, temperature moderately lowers the $S_{2n}$, and the drip-line nuclei within the systematic error are $N = 126–128$. At $T = 1.0$ MeV, nucleons have enough energy to start populating states above the closed shell, which shifts the two-neutron drip line to $N = 130–132$. At $T = 2.0$ MeV, the $S_{2n}$ isotopic dependence assumes almost a linear form, which brings more nuclei within the drip line. Now, $N = 134–138$ is the drip line accounting for systematic uncertainties. To conclude, for $Z = 60$, present relativistic EDF calculations predict that when the temperature increases from 0 to 2 MeV, the neutron number of particle-bound nuclei at finite temperature increases at least by eight and at most by twelve. Since nuclei become less bound with increasing temperature[34,38,40], one could expect that at finite temperatures the overall number of bound nuclei within the nuclide chart should become smaller. The surprising results of our theoretical study show that for high temperatures $T \gtrsim 1$ MeV, the number of nuclei within the drip lines increases, because of the thermal quenching of the shell effects.

## Impact of continuum subtraction on the location of drip lines

How important is the treatment of continuum for the description of the drip line? To demonstrate the relevance of continuum subtraction in calculations, in Fig. 2, we compare the location of the two-neutron drip line with and without the BLV continuum subtraction. The results indicate that neglecting the continuum subtraction leads to a drip line that is more neutron-rich, with 3 to 4 additional bound even-even nuclei for each isotopic chain, as depicted in Fig. 2 at $T = 1$ MeV (blue

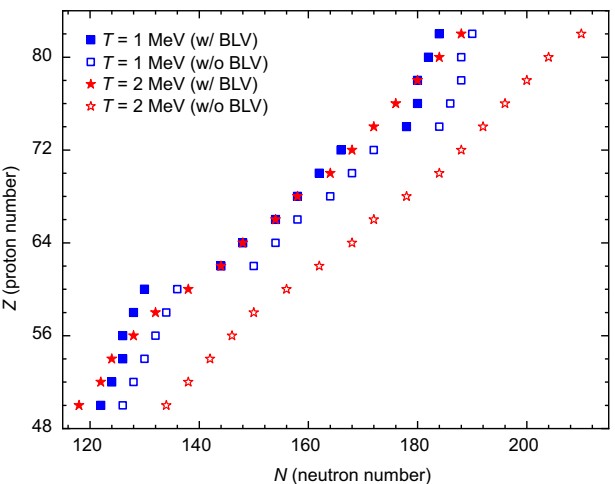

**Fig. 2 | Impact of continuum subtraction on the prediction of the two-neutron drip line.** The location of the two-neutron drip line between tin ($Z = 50$) and lead ($Z = 82$) at $T = 1$ MeV (blue squares) and $T = 2$ MeV (red stars). The calculations without the BLV continuum subtraction (empty symbols) are shown together with the ones including the continuum subtraction (filled symbols). The calculations are performed with the DD-PC1 interaction.

squares). The effects become even more pronounced with increasing temperature, as illustrated at $T = 2$ MeV (red stars), where the neutron drip line is pushed further and for each isotopic chain it would include around an additional 10 even-even nuclei without the continuum subtraction. These findings highlight the necessity of considering the particle continuum for an accurate description of the neutron drip line, particularly for $T \approx 1$ MeV and above, to prevent overestimation of the number of bound nuclei within the drip lines.

## Sensitivity of drip lines for hot nuclei to the nuclear symmetry energy

It is well established that nuclear properties near the drip lines are highly sensitive to changes in the symmetry energy parameters of the nuclear equation of state (EOS). A higher value of the symmetry energy at saturation density ($J$) is known to energetically favor more neutron-rich nuclei and expand the limits of the drip lines[49–51]. The relativistic functionals utilized in this study display similarities in terms of their predictions for the drip lines in view of the underlying symmetry energy values. To validate our findings for the drip lines at finite temperature, we performed additional calculations using the set of density-dependent point-coupling functionals DD-PCJ, which encompass a range of symmetry energy parameters with constrained values of $J$ at 30, 32, 34, and 36 MeV[52]. The results are presented in Fig. 3 for $T = 0.5$ MeV (a) and $T = 2$ MeV (b). At $T = 0.5$ MeV, we observe that shell effects are still present, as evidenced by the $N = 82$, 126, and 184 magic numbers. While the proton drip line is not sensitive to changes in the symmetry energy values, the two-neutron drip line displays a systematic dependence on the symmetry energy parameter $J$, and more nuclei are found to be bound with high $J$ values compared to those with lower $J$ values, as expected[49–51]. At $T = 2$ MeV, the shell effects disappear, and neutron drip line is described approximately by a simple straight line. We find that the DD-PCJ36 predicts around 250 additional bound even-even nuclei compared to the DD-PCJ30 at $T = 2$ MeV. Considering our findings using different EDFs, we conclude that the impact of the temperature on drip lines remains similar, regardless of the functional employed in the calculations.

In Fig. 4a, we demonstrate the variation of the number of bound nuclei $N_{nucl}$ with temperature for seven relativistic EDFs considered in this work. For a set of DD-PCJ functionals constrained to a specific symmetry energy $J$ a clear hierarchy is observed. Functionals with

lower $J$ tend to predict a smaller number of bound nuclei for all considered temperatures. Although DD-ME2 ($J = 32.3$ MeV), DD-PC1 ($J = 33$ MeV), and DD-PCX ($J = 31.1$ MeV) have distinct formulations and optimisation procedures employed to constrain their parameters, we also observe a comparable dependence on the symmetry energy values. However, a more detailed comparison requires investigating the full density-dependence of the symmetry energy $S_2$. To provide a more comprehensible representation of the variations in the number of bound nuclei as temperature increases, we present in Fig. 4b the relative change of $N_{nucl}$ with respect to zero temperature. At low temperatures, the interplay between the properties of nuclear effective interaction, pairing, and temperature effects determines the nuclear binding and number of bound nuclei, exhibiting a nearly

constant or slightly decreasing trend. As the temperature is raised to $T = 1$ MeV, the number of bound nuclei undergoes variation, either with a slight increase or decrease, depending upon the specific functional employed in the calculation. At higher temperatures, the pairing correlations disappear, shell effects vanish, and the number of bound nuclei starts to increase with increasing temperature for all functionals considered in this work.

## Temperature evolution of the quadrupole deformation

What happens to the shape of the nucleus as the temperature increases? It is well known that most nuclei at zero temperature display axially deformed shapes, with only a handful in the vicinity of closed shells being spherical[35,53]. One can imagine a nucleus as a ball rolling on a surface whose curvature is determined by its shape and eventually coming to a halt in a local minimum – illustrating the concept of the potential energy surface (PES). Assuming axial symmetry, the deformation can be quantified using the quadrupole deformation parameter $\beta_2$, where negative (positive) $\beta_2$ represents oblate (prolate) shape. To demonstrate the impact of temperature on the nuclear shape, in Fig. 5, we show the optimal quadrupole deformations for the transitional/lanthanide nuclear region determined by $\beta_2^*$ at $T = 0.5$ MeV (a) and $T = 2$ MeV (b), using DD-ME2 interaction. The panels below the nuclide maps show the subtracted neutron density with the corresponding PES, which defines the optimal deformation $\beta_2^*$. At temperature of $T = 0.5$ MeV, only the nuclei near the $N = 126$ shell closure attain spherical shapes. For instance, $^{180}$Gd is predicted to be an oblate nucleus (flattened from the top), while on the other hand, $^{210}$Gd is prolate (elongated in the direction of its polar axis). Their PESs have a rather complicated shape with pronounced minima on both the prolate and oblate sides. However, with increasing temperature at $T = 2$ MeV a dramatic change occurs, and most nuclei become spherical. The PES attains a much simpler and smoother structure. The additional energy supplied by the environment, required to keep the nucleus in excited states, results in the spherical shape of $^{180}$Gd. On the other hand, $^{210}$Gd is still deformed, however, its PES becomes significantly "flattened", reducing the energy barrier between the oblate and prolate minima. Such a result leads to the conclusion that phenomena occurring at high temperatures $T > 2$ MeV, such as in core-collapse supernovae, require only a simple spherical treatment of nuclear geometry.

## The neutron emission lifetimes at finite temperature

There is an important difference between the drip lines at zero temperature and finite temperature. At zero temperature, nuclei within the

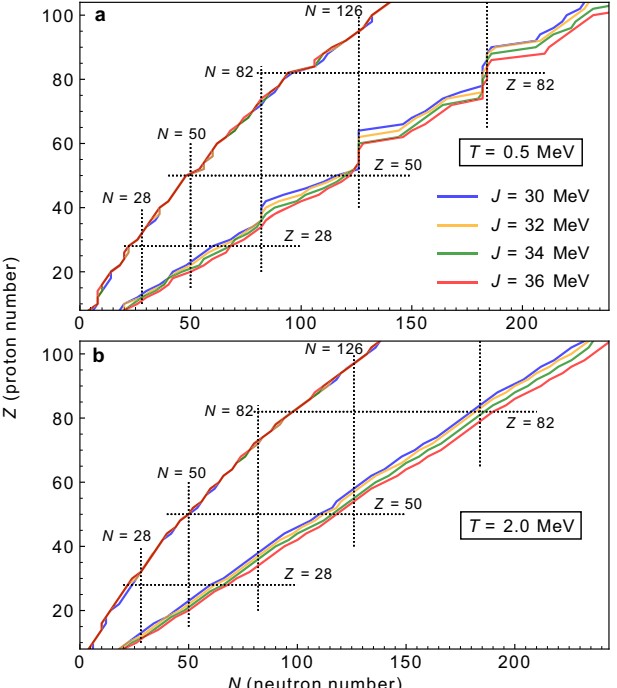

**Fig. 3 | Connection between drip lines at finite temperature and symmetry energy. a, b** The two-nucleon drip lines at $T = 0.5$ MeV and $T = 2$ MeV. The calculations are performed using the relativistic density-dependent point-coupling functionals constrained to the symmetry energy values at $J = 30, 32, 34$ and $36$ MeV. Dotted lines denote the shell-closure numbers.

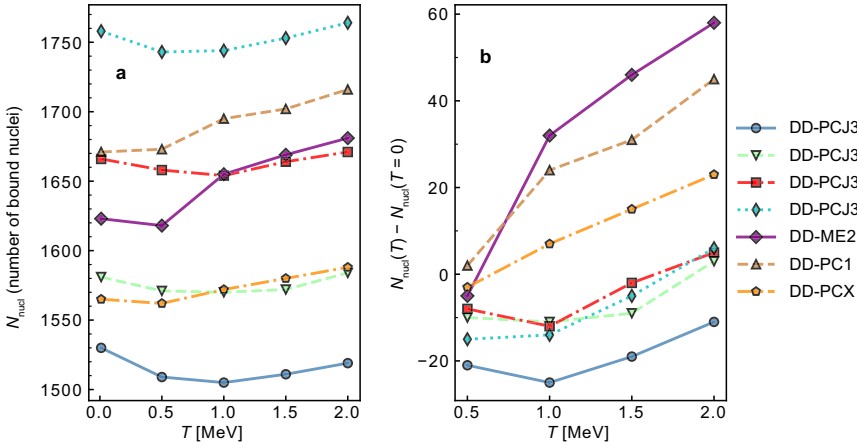

**Fig. 4 | Total number of bound nuclei. a** Temperature evolution of the total number of bound nuclei $N_{nucl}$ for seven relativistic EDFs considered in this work. **b** Relative change of $N_{nucl}$ with temperature calculated with respect to the number of nuclei at zero temperature $N_{nucl}(T = 0)$.

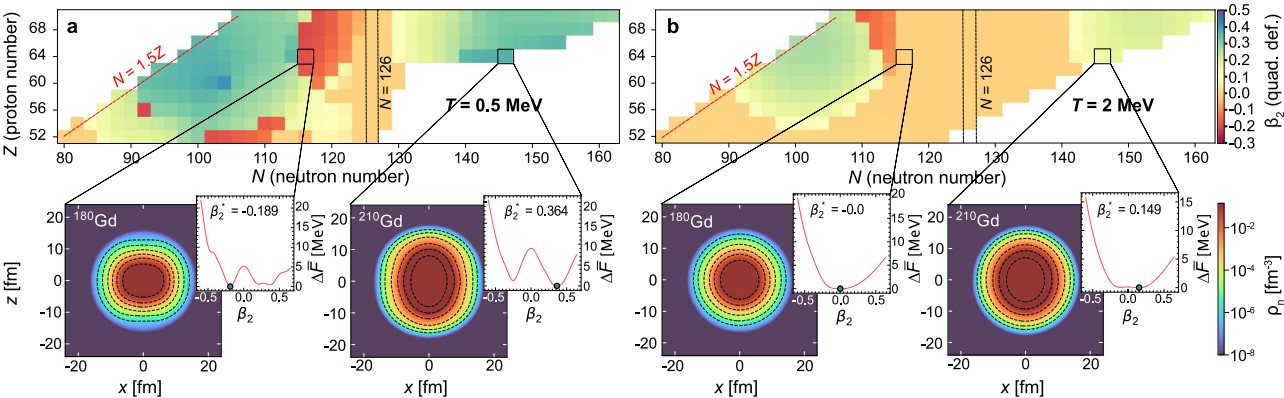

**Fig. 5 | Temperature evolution of quadrupole deformation. a, b** The isotopic dependence of the quadrupole deformation ($\beta_2$) for neutron-rich nuclei in the transition/lanthanide region at temperatures $T = 0.5$ MeV and $T = 2.0$ MeV. The continuum subtracted neutron density profiles for two gadolinium isotopes $^{180}$Gd

and $^{210}$Gd are displayed, demonstrating the optimal shapes (denoted by $\beta_2^*$) at given temperature. Insets on the right of the neutron density maps show the dependence of the relative subtracted free energy ($\Delta\bar{F}$), on the nuclear shape ($\beta_2$), with respect to minimum configuration (blue circle).

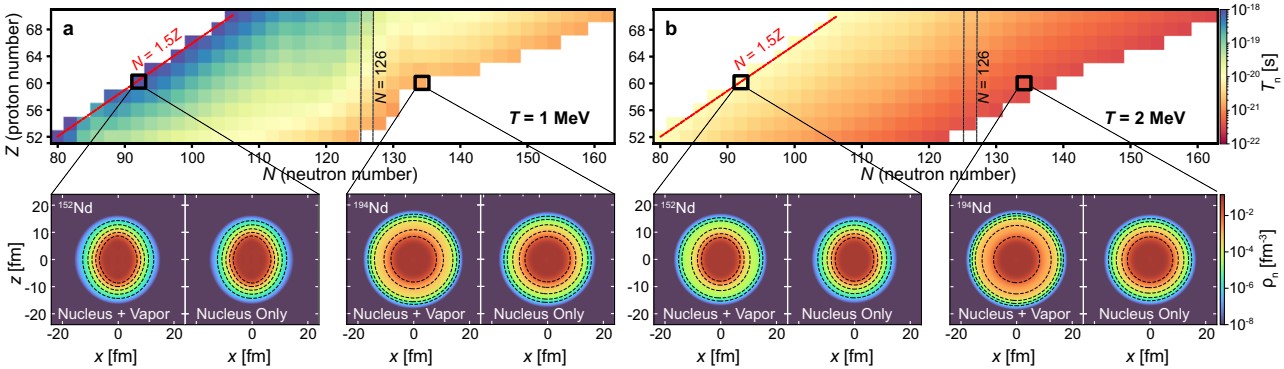

**Fig. 6 | Temperature evolution of neutron emission lifetimes. a, b** The neutron emission lifetimes ($T_n$) for neutron-rich ($N \geq 1.5Z$) nuclei in the transition/lanthanide region for temperatures $T = 1.0$ MeV and $T = 2.0$ MeV. For $^{152}$Nd and $^{194}$Nd we show the neutron density ($\rho_n$) profiles of the total unsubtracted solution (Nucleus

+Vapor) and the subtracted density profile where the nucleon vapor has been isolated (Nucleus Only). For presentation purposes, the even-even nuclei are shown side-by-side.

drip lines are stable with respect to nucleon emission (either proton or neutron). With the introduction of temperature, the nucleus within the drip lines exists in a metastable state, which possesses a certain width $\Gamma$ for particle emission[41,54]. Such decay occurs in equilibrium if the decay time ($\hbar/\Gamma$) is longer than the corresponding thermalization time. Considering a specific case of the neutron emission, in Fig. 6, we show the neutron emission lifetimes for neutron-rich nuclei in the $52 \leq Z \leq 70$ region at $T = 1$ MeV (a) and $T = 2$ MeV (b), using DD-ME2 interaction (for more details on calculation see Supplementary Information). As one approaches the neutron drip line, the lifetimes get shorter, while heating the nucleus from 1 to 2 MeV results in a significant decrease in lifetimes as well. To further exemplify such a result and the importance of proper continuum subtraction at finite temperature, we choose an example of two neodymium nuclei— $^{152}$Nd and $^{194}$Nd, and show their neutron density profiles in lower panels of Fig. 6a–b. By employing the proper continuum subtraction procedure, we can isolate the nucleon vapor contribution due to continuum states from the pure nuclear contribution. In the case of $^{152}$Nd at $T = 1$ MeV, vapor contribution is small, hence the two solutions are similar. However, for $^{194}$Nd located at the neutron drip line, the vapor alters the neutron density distribution, especially in the tail region. On the other hand, at $T = 2$ MeV, the nucleon vapor has a large contribution in both $^{152}$Nd and $^{194}$Nd. The neutron emission lifetimes are proportional to the nucleon vapor density[42–44,54], and hence decrease towards the neutron drip line as well as with increasing temperature.

## Discussion

We note that our calculations do not incorporate approaches beyond the mean field or account for statistical (thermal) fluctuations. The effects of the beyond-mean-field approaches in the predictions of pairing gaps and single-particle spectrum have been discussed in Refs. 55–59 at zero temperature, and it has been shown that the impact of beyond-mean-field phenomena on pairing gaps is significant, resulting in a notable state-dependent variation of the gaps and an increase in the pairing gaps near the Fermi surface. Incorporating beyond-mean-field approaches in the calculations at finite temperatures leads to the fragmentation of the single-particle spectrum and is also expected to result in higher critical temperatures ($T_c > 1.0$ MeV) for pairing phase transitions in nuclei[39,60,61]. While the impact of beyond mean field approaches can be important for the calculations of drip lines at low temperatures ($T < 1$ MeV)[59], we anticipate that its effect will be minor at high temperatures due to the quenching of shell effects[61]. For a realistic description of nuclei at finite temperatures, the inclusion of statistical or thermal fluctuations is also necessary. By taking into account the thermal fluctuations, a smoother decrease is expected in pairing gap and deformation properties as temperature increases rather than a sharp decrease in these properties[15,37,62–65]. Therefore, it is expected that the pairing gap and deformation properties can persist, albeit small, at high temperatures and above $T > 1$ MeV when thermal fluctuations are taken into account. However, performing large-scale calculations utilizing these techniques is not feasible at present due to

their high computational demands. Furthermore, in determining the two-neutron drip lines, we rely on the subtraction of the free energies, which further mitigates the impact of these correlations.

Considering the importance of shell closures for nuclear properties, it remains an open question how thermal shell quenching and shown quantification of the increased number of nuclei within the drip lines could impact the modelling of extreme astrophysical events such as neutron star mergers and core-collapse supernovae. With the advent of the era of multimessenger astronomy, understanding nuclear properties at high-temperature sources of new signals from the Universe becomes essential for disclosing their origin. It is necessary to perform the relevant astrophysical studies by treating thermalized nuclei as being in a metastable state. This implies an interpretation of the nuclear drip line not as a "fixed" limit of the nuclear landscape but rather as a dynamically changing limit depending on temperature.

## Methods

Theoretical framework employed in this work includes the relativistic Hartree-Bogoliubov (RHB) model in the axially-deformed harmonic oscillator basis[66,67], which is additionally extended to include finite temperature (FT-RHB). To treat the continuum states, we use the BLV approach[42,43], which makes our finite temperature results independent of the box size (in our case, the number of harmonic oscillator shells $N_{osc}$ and the oscillator length $b_0$). The BLV procedure consists of solving the two coupled FT-RHB equations, one for the nucleus and vapor system (N+V) and the other for vapor (V) only. The resulting coupled FT-RHB equations are solved through self-consistent iterations by utilizing the modified Broyden mixing[68]. Before starting the self-consistent procedure the mean- and pairing-field should be initialized. The N+V system follows the usual prescription as in Ref. 68, where the nuclear mean-field assumes a combination of the Woods-Saxon and the Coulomb potential. On the other hand, the V system is only initialized by the Coulomb potential. The total density of the nucleus is calculated as a difference between the N+V density $\rho_{N+V}$ and V density $\rho_V$ as $\bar{\rho} = \rho_{N+V} - \rho_V$. All observables of interest are then expressed as a function of the subtracted density $\mathcal{O}[\bar{\rho}]$. Such a procedure guarantees the independence of observables on the basis parameters such as $N_{osc}$ (details are presented in Supplementary Note 1). In this work, we have developed a parallelized version of the axially-deformed code DIRHBz[68] extended to finite temperature (FT-DIRHBz), to solve coupled FT-RHB equations for N+V and V systems. The calculations are performed for seven relativistic EDFs that can be grouped into two main categories. One is the density-dependent meson-exchange effective interaction, DD-ME2[46], and others include point-coupling functionals, DD-PC1[47] and DD-PCX[48], in addition to 4 functionals from the DD-PCJ family that vary the symmetry energy at saturation density ($J$)[52]. Considering different functionals provides a convenient method to study systematic variations between different EDFs. These functionals were established using rather different optimization protocols to constrain their parameters: for DD-ME2 a small set of bulk nuclear properties was used[46], for DD-PC1 only binding energies for deformed nuclei were used[47], and to adjust DD-PCX in addition to nuclear ground state properties, also nuclear excitations have been considered[48]. The DD-PCJ family is established by using an additional constraint by the symmetry energy $J$ value, and we chose functionals with $J$ = 30, 32, 34 and 36 MeV[52]. The data for temperature-dependent drip lines using all considered functionals is deposited in Ref. 69. Such different functionals lead to a considerable variety in the particle-hole channel of the nuclear mean-field in addition to the isovector dependence of EDFs through symmetry energy. In the particle-particle channel, separable pairing interaction is used[70] for all employed functionals, with interaction strength constants $G$ differing amongst EDFs (see Supplementary Note 1C). The FT-RHB equations are solved by expanding the wavefunctions and meson fields in the basis of axially-deformed harmonic oscillator characterized by $N_{osc}$ shells. Both fermion (nucleon

Dirac spinors) and boson (minimal set of $\sigma$, $\omega$ and $\rho$ mesons appearing only in the DD-ME2 EDF) states are expanded in $N_{osc}$ = 20, which yields satisfying convergence properties for our finite temperature study. For each nucleus, we perform constrained calculations for 11 points on a quadrupole deformation mesh, from $\beta_2 = -0.6$ to $\beta_2 = 0.7$. The constraint is kept for the first 20 iterations, during which the potential energy surface (PES) is stabilized, and then the constraint is removed and unconstrained calculations are performed until convergence is achieved. This procedure usually yields two minima with prolate and oblate deformations. We store the state which minimizes the subtracted free energy $\bar{F}$ for further calculation. The neutron emission lifetimes are calculated by following the procedure outlined in Refs. 42,71, with detailed description in Supplementary Note 3.

## Data availability

The finite-temperature drip line data generated in this study have been deposited in the Figshare database under accession code https://doi.org/10.6084/m9.figshare.23671839.v1.

## Code availability

The FT-DIRHB code with the BLV continuum subtraction procedure employed to calculate the drip lines is available at https://github.com/aravlic/DIRHBZ_BLV.git, along with the minimal working example.

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

## Acknowledgements

We are grateful to W. Nazarewicz and S. E. Agbemava for helpful discussions. This work is supported by the QuantiXLie Centre of Excellence, a project co financed by the Croatian Government and European Union through the European Regional Development Fund, the Competitiveness and Cohesion Operational Programme (KK.01.1.1.01.0004) (A.R., T.N., E.Y., N.P.). Support from the Science and Technology Facilities Council (UK) through grant ST/Y000013/1 is also acknowledged (E.Y.). This work was supported in part through computational resources and services provided by the Institute for Cyber-Enabled Research at Michigan State University (A.R.). We acknowledge support by the US National Science Foundation under Grant PHY-1927130 (AccelNet-WOU: International Research Network for Nuclear Astrophysics [IReNA]) (A.R., N.P.).

## Author contributions

Theoretical calculations were performed by A.R. and E.Y. The data were analyzed by A.R., E.Y., T.N. and N.P. The manuscript was prepared by A.R., E.Y., T.N. and N.P. All authors contributed to this work, discussed results and conclusions, and commented on the manuscript.

## Competing interests

The authors declare no competing interests.
