## [Peer Review File · Nature Communications]

Expanding the limits of nuclear stability at finite temperatureREVIEWER COMMENTS

Reviewer #1 (Remarks to the Author):

The paper presents the results of extensive calculations of nuclei throughout the mass table using density functional theory at finite temperature. The focus is on the dependence of the limits of nuclear stability on temperature.

The theoretical framework is well established, and the paper is not particularly innovative in this respect. On the other hand, the manuscript is clearly written and the subject is of high interest for the nuclear research community, and is appropriate for Nature Communications. The calculations represent a significant computational effort and an advance for the field; the list of drip lines provided in the Supplemental Material can be an important benchmark for future calculations.

One could argue, whether the results are relevant enough to be published in Nature Communications.

In my opinion the paper might be acceptable for publication, after a number of improvements have been introduced, as indicated below.

- The authors state that they 'report the first mapping of drip lines for hot nuclei'. I find it puzzling that a recent paper by one of the authors on the same subject is not even quoted (E. Yuksel, Nucl. Phys. A 1014 (2021) 122328). This paper deals with the same topic, and uses essentially the same theoretical approach, although with relevant differences (a non relativistic functional is adopted, and no continuum subtraction is implemented). In particular, two-neutron separation energies up to the neutron drip lines are presented (see fig. 8 in that paper). I think that such differences and possibly the improvements compared to the previous paper should be carefully discussed. Furthermore, a comparison between the results obtained with relativistic and with nonrelativistic functionals would be welcome. In particular, Fig. 8 shows important differences in the location of the drip lines adopting a criterion based on the two-neutron separation energies or on the neutron chemical potential. This point seems to differ from the results reported in Section B of the Supplemental material of the present paper.

- The authors do not discuss the limitations of their approach. They should instead carefully consider the qualitative changes that may occur to their picture, by introducing effects beyond mean field. The effects of shape and pairing fluctuations at finite temperature are quite important have been discussed

many times and in various contexts, although probably not in the specific case of the location of drip lines. Appropriate references should be given. The increase in the number of bound nuclei in the region near closed shells seems to be due to the smoothing of the occupation profile around the Fermi surface at temperatures larger than 1 MeV, but what would occur taking into account correlations that would smear such profile already at zero temperature?

- In several places, consequences of the present work related to astrophysical processes are mentioned. In my opinion, these statements are rather vague and I do not think that they deserve to be included, unless the related effects can be precisely identified.

A sentence like 'the determination of the nuclear properties and drip lines at finite temperature is essential' for example, mixes up a general and obvious statement (the determination of the nuclear properties at finite temperature is essential) with the specific and problem at hand: the determination of the drip lines is essential. Is the latter sentence well justified?

- The continuum subtraction method is mentioned very rapidly in the Method Summary. I think that it should be summarised more extensively and clearly in the supplemental material. Also, no explanation is given, about the calculation of lifetimes reported in Fig. 3.

- I think that the abstract should be rewritten. It is a collage of several sentences, not well linked together, and

partly obvious or rather superficial.

Reviewer #2 (Remarks to the Author):

Review of "Expanding the limits of nuclear stability: Drip lines for hot nuclei"

The authors studied the properties finite nuclei at finite temperature which might be useful for better understanding of inhomogeneous nuclear matter in the core-collapsing supernovae environment.

They used the relativistic DFT with recent parameter sets which are quite consistent with recent development or knowledge of nuclear matter properties (symmetric nuclear matter, and S_v, L parameters).

I don't think, however, it is not appropriate on the current levels of work to publish in this journal based on the review criteria.

First of all, in the point of view of nuclear astrophysics, or supernova EOS, the dripline is not quite useful because the simulation needs the EOSs which give information of finite nuclei(average atomic number and average mass number), and dripped neutrons and protons.

This paper only deals with driplines at finite temperature by considering two neutron separation energies or neutron chemical potentials.

Question : The separation energy was calculated by comparing the free energy of finite nuclei, and didn't consider the contribution of vapors, right?

I'm not quite sure this is actually correct because as the temperature increases, there are more neutrons and protons outside finite nuclei.

This paper claims that neutrons and protons(in this paper, vapors) are separated using the Bonche-Levit-Vautherin(BLV) method.

In the supernova EOS or nuclear pasta phase(inhomogenous nuclear matter), the properties finite nuclei and vapors highly depends on the total number density in the boxes. But in this paper, they didn't mention what the total baryon number density in the numerical box for the calculation is.

They mentioned that

"If the boundary conditions of continuum states are not properly treated, one finds that the results of the calculations are dependent upon the box size used to discretize the problem."

-> I agree with this point but more serious approach is necessary.

Second of all, they used DD-ME2, DD-PC1, DD-PCX.

I think they made a good choice for the EDFs in this work.

As I mentioned, they are quite consistent with the well known results for symmetric nuclear matter, and symmetry energy parameters.

I think that the amount of works with three different models is enormously huge
(for the calculation of driplines or mass tables)

but three different models -which are the almost central point of nuclear matter properties
won't be enough to generalized your results.

Calculations from some models with different nuclear matter properties might make your work solid.

In the same matter, the study of different temperature is limited to

$T=0.0\text{MeV}$, 0.5MeV , 1.0MeV , and 2.0MeV .

More temperature grid points are necessary to generalized your results.

To support your arguments such as,

"At $T = 2.0 \text{ MeV}$, shell effects are

137 completely washed out, nuclei are mostly spherical, and the neutron drip line

138 is described by a simple straight line. "

or

"However, with increasing tem-

197 perature at $T = 2 \text{ MeV}$ a dramatic change occurs, and most nuclei become

198 spherical."

I think you need more grid points between 1MeV and 2MeV.

I agree with your observation that the deformation vanishes as temperature increases.

I'm also wondering if you can show the plots (β_2 - yaxis, Temperature-xaxis) to support your results.

In the supplementary materials,

your paper says

"DD-PCX tends to predict the two-neutron dripline at a lower number of neutron N."

Why does it happen?

Was it caused by the difference from the different pairing parameters?

What kind of pairing models did you use in this work?

If there is a specific pairing model for each EDF, why is there specific model?

If not, why?

Maybe same question,

"If N_{nucl} denotes the number of even-even nuclei between the drip lines,

then it can be inferred that $N_{\text{nucl}}(\text{DD-PC1}) > N_{\text{nucl}}(\text{DD-ME2}) > N_{\text{nucl}}(\text{DD-PCX})$, for all studied temperatures."

What is the fundamental reason for this?

In your numerical work, the harmonic oscillator basis(HO) is used.

But I'm wondering if it is adequate when there exist vapors.

For example, HO is used for bound nuclei and if "HO" is used, most nuclei is more like tightly bound.

Thus to study halo nuclei, the transformed harmonic oscillator(THO) basis gives better results.

How can you justify your results using HO basis?

Reviewer #3 (Remarks to the Author):

I find this a very interesting piece of work worth of being considered for publication. I would support publication as it is. The article is well written and to the point with several interesting physics messages about drippiness and the influence of temperature, with great consequences for the modeling of the synthesis of nuclei.

REPLY TO THE REVIEWERS' COMMENTS ON MANUSCRIPT

NCOMMS-22-49107

We thank the reviewers for their careful reading of our paper, thoughtful and useful comments and suggestions. By considering the reviewers' questions and comments, we revised and improved the manuscript. We made significant modifications and extensions both in the main article manuscript and the supplementary material. Furthermore, to respond to the reviewers' comments, we extended our study of the drip lines by introducing four additional relativistic EDFs, we doubled the number of temperature mesh points, and performed a significant amount of new calculations, accordingly. Following reviewers' comments and 'Guide for submission to Nature Communications', we have also rewritten the Abstract and the Introduction. In the following, we reply to all the questions and describe the modifications made in the main article and Supplementary Information (highlighted in text by red color).

Reviewer #1

1) *The authors state that they 'report the first mapping of drip lines for hot nuclei'. I find it puzzling that a recent paper by one of the authors on the same subject is not even quoted (E. Yuksel, Nucl. Phys. A 1014 (2021) 122328). This paper deals with the same topic, and uses essentially the same theoretical approach, although with relevant differences (a non relativistic functional is adopted, and no continuum subtraction is implemented). In particular, two-neutron separation energies up to the neutron drip lines are presented (see fig. 8 in that paper). I think that such differences and possibly the improvements compared to the previous paper should be carefully discussed. Furthermore, a comparison between the results obtained with relativistic and with nonrelativistic functionals would be welcome. In particular, Fig. 8 shows important differences in the location of the drip lines adopting a criterion based on the two-neutron separation energies or on the neutron chemical potential. This point seems to differ from the results reported in Section B of the Supplemental material of the present paper.*

We have included Ref. [1] in the revised version of the manuscript, and in the Supplementary Information we have compared the results obtained with relativistic and with

non-relativistic functionals from Ref. [1] (See Supplementary Note ID). As pointed out by the Referee, in Ref. [1], the Skyrme-type SkM* interaction was used, and the continuum subtraction method was not implemented in the calculations. Thus, the previous investigation was not reliable because of the deficiencies in the model, and two-neutron drip lines were predicted at considerably higher neutron numbers, as we demonstrate in the revised version of our work.

First, we have compared the results obtained by using the DD-PC1 functional with those from Ref. [1] obtained by using the SkM* functional without employing the BLV method in order to ensure consistency in the calculation procedure (see Supplementary Fig. 1 and explanation on Pages 7-9 in the Supplementary material of the revised manuscript). We have found that both the relativistic and non-relativistic approaches exhibit the same behaviour of the drip line with increasing neutron number and temperature. Furthermore, in both cases the drip lines definitions based on the chemical potential and two-neutron separation energies are incompatible when the temperature is increased. This problem is resolved by implementing the continuum subtraction method, as we have done in our work. We have presented our results obtained by using the FT-RHB+BLV method to demonstrate that inclusion of the continuum subtraction method is crucial for accurately describing weakly bound nuclei near the drip lines at finite temperatures (see Supplementary Note IIC). Our results show that drip line definitions based on chemical potential and two-neutron separation energies are compatible only with the inclusion of the BLV procedure at finite temperatures (see Supplementary Figure 7 and related explanation on pages 15–16 in Supplementary Information).

To investigate the influence of the BLV continuum subtraction procedure on the prediction of the drip lines, we have added Fig. 2 in the updated version of the main manuscript. The figure demonstrates that including the BLV subtraction is crucial for determining the neutron drip line at a finite temperature. Results for $T \gtrsim 1$ MeV without the continuum subtraction do not only predict more neutron-rich drip lines but are also dependent on the underlying basis (e.g., HO shells). We have added the corresponding discussion starting from line 189 of the updated version of the main manuscript:

” How important is the treatment of continuum for the description of the drip line? To demonstrate the relevance of continuum subtraction in calculations, in Fig. 2, we compare

the location of the two-neutron drip line with and without the BLV continuum subtraction. The results indicate that neglecting the continuum subtraction leads to a drip line that is more neutron-rich, with 3 to 4 additional bound even-even nuclei for each isotopic chain, as depicted in Fig. 2(a) at $T = 1$ MeV. The effects become even more pronounced with increasing temperature, as illustrated in Fig. 2(b) at $T = 2$ MeV, where the neutron drip line is pushed further and for each isotope chain it would include around an additional 10 even-even nuclei without the continuum subtraction. These findings highlight the necessity of considering the particle continuum for an accurate description of the neutron drip line, particularly for $T \approx 1$ MeV and above, to prevent overestimation of the number of bound nuclei within the drip lines. ”

We note that incorporating the BLV method in the Skyrme-HFBTHO code poses a considerable challenge, considering that for most of the authors in this work the major in-depth expertise is in the relativistic EDFs. The effort and required time to establish the non-relativistic model with the BLV method included, numerical implementation, tests of the codes, etc. go significantly beyond our timeline available for this submission. Therefore, we have opted not to integrate the BLV method into the HFBTHO code. Nonetheless, based on the comparable outcomes obtained using both relativistic and non-relativistic functionals without the BLV method, and usage of the same theoretical method in the calculations, we anticipate that the Skyrme-type functionals will display similar behaviour if the BLV method is included.

We also note that in the previous studies of drip-lines, those limited at zero temperature, publications mainly include either relativistic or nonrelativistic approaches, depending on the major expertise of the research groups, e.g., J. Erler et al., *Nature* 486, 509–512 (2012); W. Nazarewicz, *Nature Physics* 14, 537–541 (2018); A. V. Afanasjev et al., *Physics Letters B* 726, 628 (2013); etc.

2) *The authors do not discuss the limitations of their approach. They should instead carefully consider the qualitative changes that may occur to their picture, by introducing effects beyond mean field. The effects of shape and pairing fluctuations at finite temperature are quite important have been discussed many times and in various contexts, although probably not in the specific case of the location of drip lines. Appropriate references should be*

given. The increase in the number of bound nuclei in the region near closed shells seems to be due to the smoothing of the occupation profile around the Fermi surface at temperatures larger than 1 MeV, but what would occur taking into account correlations that would smear such profile already at zero temperature?

We thank to the referee for pointing out these issues. We also agree that inclusion of the beyond mean field approaches and thermal fluctuations are expected to impact the results by increasing the critical temperatures for the pairing phase transition and deformation properties of nuclei. On pages 16–17 in the updated version of the main manuscript, we discuss the possible impact of the inclusion of the beyond mean field approaches and thermal fluctuations in the calculations with the relevant references:

”We note that our calculations do not incorporate approaches beyond the mean field or account for statistical (thermal) fluctuations. The effects of the beyond-mean-field approaches in the predictions of pairing gaps and single-particle spectrum have been discussed in Refs. [2–5] at zero temperature, and it has been shown that the impact of beyond-mean-field phenomena on pairing gaps is significant, resulting in a notable state-dependent variation of the gaps and an increase in the pairing gaps near the Fermi surface. Incorporating beyond-mean field approaches in the calculations at finite temperatures leads to the fragmentation of the single-particle spectrum and is also expected to result in higher critical temperatures ($T_c > 1.0$ MeV) for pairing phase transitions in nuclei [6–8]. While the impact of beyond mean field approaches can be important for the calculations of drip lines at low temperatures ($T < 1$ MeV), we anticipate that its effect will be minor at high temperatures due to the quenching of shell effects [8]. For a realistic description of nuclei at finite temperatures, the inclusion of statistical or thermal fluctuations is also necessary. By taking into account the thermal fluctuations, a smoother decrease is expected in pairing gap and deformation properties as temperature increases rather than a sharp decrease in these properties [9–14]. Therefore, it is expected that the pairing gap and deformation properties can persist, albeit small, at high temperatures and above $T > 1$ MeV when thermal fluctuations are taken into account. However, performing large-scale calculations utilizing these techniques is not feasible at present due to their high computational demands. Furthermore, in determining the two-neutron drip lines, we rely on the subtraction of the free energies, which further mitigates the impact of these correlations. Thus, the inclusion of both beyond-mean

field approaches and thermal fluctuations in the calculations can lead to slight changes in the predictions of the drip lines at zero and finite temperatures, and our findings concerning the mapping of the drip lines will retain their validity, especially at higher temperatures.”

3) *In several places, consequences of the present work related to astrophysical processes are mentioned. In my opinion, these statements are rather vague and I do not think that they deserve to be included, unless the related effects can be precisely identified. A sentence like ‘the determination of the nuclear properties and drip lines at finite temperature is essential’ for example, mixes up a general and obvious statement (the determination of the nuclear properties at finite temperature is essential) with the specific and problem at hand: the determination of the drip lines is essential. Is the latter sentence well justified?*

We appreciate the referee’s feedback. We softened our arguments about the impact of our findings on astrophysical processes in the manuscript because this mainly relates to investigations that have to be performed in forthcoming studies. We made the following changes in the manuscript:

- The abstract is rewritten, and we focused more on the changes in the drip lines rather than its impact in nuclear astrophysical processes.
- We modified the Introduction on pages 3–5,
- We removed the sentence ”‘the determination of the nuclear properties and drip lines at finite temperature is essential’ to avoid the confusion.
- We modified the sentence in the discussion, lines 312-316

We hope that these modifications will address the referee’s concerns and improve the clarity of our manuscript.

4) *The continuum subtraction method is mentioned very rapidly in the Method Summary. I think that it should be summarised more extensively and clearly in the supplemental material. Also, no explanation is given, about the calculation of lifetimes reported in Fig. 3.*

Based on the referee’s comment, we have expanded our discussion of the continuum sub-

traction method in the Supplementary Note IA–C to provide more comprehensive and clear explanations. Additionally, we have included details about the calculation of the neutron emission lifetimes in Supplementary Note III. The discussion in the Methods section of the main manuscript has been broadened as well. We hope that these additions will address the referee’s concerns and improve the clarity and completeness of our manuscript.

5) *I think that the abstract should be rewritten. It is a collage of several sentences, not well linked together, and partly obvious or rather superficial.*

We have rewritten the abstract in response to the Referee’s comments, also following the ‘Guide for submission to Nature Communications’, that requires for the abstract “a general introduction to the topic and a brief non- technical summary of your main results and their implication”:

”Properties of nuclei in hot stellar environments such as supernovae or neutron star mergers are largely unexplored. Since it is poorly understood how many protons and neutrons can be bound together in hot nuclei, we investigate the limits of nuclear existence (drip lines) at finite temperature. Here, we present mapping of nuclear drip lines at temperatures up to around 20 billion kelvins using the relativistic energy density functional theory (REDF) with essential treatment of thermal scattering of nucleons in the continuum. With extensive computational effort, the drip lines are determined using several REDFs with different underlying interactions, demonstrating considerable alterations of the neutron drip line with temperature increase, especially near the magic numbers. We find a surprising result that the total number of bound nuclei is larger at high temperatures compared to cold nuclei, due to thermal shell quenching. Our findings provide insight into nuclear landscape for hot nuclei, revealing that the nuclear drip lines should be viewed as limits that change dynamically with temperature.”

Reviewer #2

1) *First of all, in the point of view of nuclear astrophysics, or supernova EOS, the dripline is not quite useful because the simulation needs the EOSs which give information of finite*

nuclei(average atomic number and average mass number), and dripped neutrons and protons. This paper only deals with driplines at finite temperature by considering two neutron separation energies or neutron chemical potentials.

We thank the referee for his/her comments. In this manuscript, we are concerned with the influence of temperature on the properties of nuclei and the drip lines. The EoS of stellar matter at finite temperature is beyond the scope of the current work. It is known that the changes in the EoS can result in changes in the neutron/proton ratios that are allowed in bound nuclei, which in turn can affect the drip line properties. However, we hope that by extending our study by using several different relativistic functionals (7), we could provide insight into how different EoSs impact the predictions of the drip line at finite temperature. This is especially connected with the referee's point **5**) about DD-ME2, DD-PC1 and DD-PCX being similar enough in their properties. To remedy this, we have calculated the drip lines using an additional set of functionals that systematically vary the symmetry energy at saturation density J , and in this way we demonstrated a clear effect on the neutron drip line already at zero temperature. We also show that the effect of the temperature on the drip lines is the same using different relativistic functionals with the corresponding different EoSs. Since the EoS and the drip line properties are related to each other, changes in the drip line positions with increasing temperature can also be related to changes in the EoS, which paves the way for further studies in this field.

Secondly, in calculating the rate tables for supernovae on weak-interaction rates, it is important to know when to terminate the calculations, and that is limited by the drip line. As it changes with increasing temperature, it would be interesting to investigate if there are any effects on the supernova observables. Of course, in practical calculations, the range of nuclei for weak-interaction rates depends on the NSE (nuclear statistical equilibrium), which in turn depends on the EoS, so the picture is more complicated.

To summarize, we made the following changes in the manuscript:

- Modifications about our extended study that relates the symmetry energy of the EoS with the drip lines are described below in reply to the point **5**).
- We modified the Introduction on page 3 of the main manuscript, including statements related to (nuclear) astrophysics

- We modified sentence in concluding part on page 17 of the updated manuscript,

”Considering the importance of shell closures for nuclear properties, it remains an open question how thermal shell quenching and shown quantification of the increased number of nuclei within the drip lines could impact the modelling of extreme astrophysical events such as neutron star mergers and core-collapse supernovae.”

2) *This paper only deals with driplines at finite temperature by considering two neutron separation energies or neutron chemical potentials. Question : The separation energy was calculated by comparing the free energy of finite nuclei, and didn't consider the contribution of vapors, right? I'm not quite sure this is actually correct because as the temperature increases, there are more neutrons and protons outside finite nuclei.*

The separation energy is calculated by considering the subtracted free-energy, which we denote as \bar{F} . It is very important to make a distinction between the subtracted free-energy calculated within the BLV procedure, compared to F . The subtracted free-energy is defined as

$$\bar{F} = F_{Nucl+Vap} - F_{Vap}, \quad (S1)$$

where $F_{Nucl+Vap}$ is the free-energy of the combined system of nucleus and vapor, while F_{Vap} is the free-energy of the vapor system only. To explain the details behind the BLV procedure, we have included the Theoretical formalism as Supplementary Note I in the updated Supplementary Information, where we start by introducing the notion of two systems at the FT-RMF level, and later expand to account for nuclear pairing within the FT-RHB. Therefore, within the BLV subtraction procedure we are solving for two systems

S 1: consisting of nucleus + vapor (Nucl + Vap), which is very similar to solving the FT-RHB equations

S 2: consisting of the vapor contribution only (Vap). This system corresponds to solving the FT-RHB equations, where the mean field Hamiltonian is initialized only with the kinetic and Coulomb parts.

Indeed, as the temperature increases, nucleons get scattered above the Fermi level, and eventually, they can have non-vanishing occupation in the particle continuum. Those states

are what we denote as the nucleon vapor, and correspond to a tail in the vector density (baryon density) as depicted in Supplementary Figure 3 of the revised Supplementary Information. It is very important to note that this vapor is artificial and consequence of the improper treatment of continuum within the FT-RHB. The contribution of the unphysical tail region in the density increases as the temperature is increased. In Supplementary Figure 2 of the updated Supp. Inf. we consider $T = 1$ MeV, which turns out to be sufficiently large for a weakly-bound ^{202}Sm . As referee remarked, neglecting the vapor contribution would lead to serious convergence issues. Both the entropy and RMS neutron radius for ^{202}Sm at $T = 1$ MeV significantly depend on the basis size (box radius R_{box} in Supplementary Fig. 2).

We summarize the changes made in the manuscript:

- introduced section on theoretical formalism in Supplementary Note I
- In Supplementary Note IIA we discuss the convergence problems without the BLV procedure, and how the BLV procedure remedies the issue. Comparison is shown in Supplementary Fig. 2. We also add Supplementary Fig. 3 which shows the radial dependence of the density and the tail due to the vapor states.
- Starting at line 109 of the main manuscript we have modified the sentence

”Therefore, at finite temperature, the two-neutron and two-proton drip lines are defined as $S_{2n} = F(Z, N) - F(Z, N - 2) \geq 0$ and $S_{2p} = F(Z, N) - F(Z - 2, N) \geq 0$, respectively.”

to:

”Within the BLV method, we consider the subtracted free-energy \bar{F} , which is devoid of the nucleon vapor contribution, and defined as $\bar{F}(Z, N) = \bar{E}(Z, N) - T\bar{S}(Z, N)$, where \bar{E} and \bar{S} are the subtracted total binding energy and entropy. Therefore, at finite temperature, the two-neutron and two-proton drip lines are defined with $S_{2n} = \bar{F}(Z, N) - \bar{F}(Z, N - 2) \geq 0$ and $S_{2p} = \bar{F}(Z, N) - \bar{F}(Z - 2, N) \geq 0$, respectively.”

We hope that these additions will address the referee’s concerns and improve the clarity and completeness of our manuscript.

3) *This paper claims that neutrons and protons(in this paper, vapors) are separated using the Bonche-Levit-Vautherin(BLV) method. In the supernova EOS or nuclear pasta phase(inhomogenous nuclear matter), the properties finite nuclei and vapors highly depends on the total number density in the boxes. But in this paper, they didn’t mention what the total baryon number density in the numerical box for the calculation is.*

It is important to note, that unlike the supernova EOS or nuclear pasta phase, the baryon density of finite-nuclei is determined by the total particle number N_q corresponding to protons ($q = 1$) and neutrons ($q = 2$). Our calculations deal with the properties of nuclei around the saturation densities $\rho_0 \approx 0.16 \text{ fm}^{-3}$.

Within the relativistic mean-field theory considered in this work, the baryon density is the vector density ρ_v . Within the BLV, we perform the variational principle not on ρ_v , but rather on the subtracted vector density $\bar{\rho}_v$ defined as

$$\bar{\rho}_v = \rho_v^{N_{ucl}+V_{ap}} - \rho_v^{V_{ap}}, \quad (\text{S2})$$

where $\rho_v^{N_{ucl}+V_{ap}}$ is the vector density of the Nuclear+Vapor system and $\rho_v^{V_{ap}}$ is the vapor density. The particle number is obtained by integrating the vector density

$$\int d\mathbf{r} \bar{\rho}_v(\mathbf{r}) = N_q, \quad (\text{S3})$$

which determines the chemical potential λ_q .

4) *They mentioned that "If the boundary conditions of continuum states are not properly treated, one finds that the results of the calculations are dependent upon the box size used to discretize the problem." - > I agree with this point but more serious approach is necessary.*

The BLV method used to treat the particle continuum within this work is a suitable approach that can be used in combination with the nuclear density functional theory (DFT). First of all, it is based on a variational method that can be naturally incorporated within the Kohn-Sham variational equations, either relativistic (as considered in this work) or

non-relativistic. The idea of separating the nuclear density in the contribution of the vapor, and a combined system of a nucleus and vapor has its microscopic justification. The fully microscopic method to treat the particle continuum would include the Green's functions methods as in Refs. [15–17], where discrete single-particle energies appear as poles and continuum states as branch cuts. Furthermore, the Green's functions have to satisfy the proper boundary conditions. However, those calculations are significantly more complicated and a systematic calculation of the nuclear drip lines would be extremely challenging (if currently possible at all). Instead, one could try to redefine the main gist of the Green's function method as in Refs. [18, 19]. The total Green function can be decomposed into two parts. One is the $\hat{G} = (\varepsilon - \hat{h} + i\eta)^{-1}$, being the causal Green function corresponding to the full single-particle Hamiltonian \hat{h} , with ε being the single particle energies and $\eta > 0$ a small parameters determining the causal structure. The other part is $G^{free} = (\varepsilon - \hat{h}^0 + i\eta)^{-1}$, where now \hat{h}^0 is the free-particle Hamiltonian consisting of the kinetic and Coulomb term only. The total Green function is defined as

$$\bar{G}(\varepsilon) = G(\varepsilon) - G^{free}(\varepsilon), \quad (\text{S4})$$

which has precisely that form as the variational density ansatz within the BLV approach [cf. Eq. (S2)]. Namely, G^{free} is the contribution of the nuclear vapor.

Therefore, the main benefits of the BLV method can be summarized as follows

- a self-consistent variational method that explicitly accounts for nuclear continuum contribution, treated as vapor
- can be incorporated into the nuclear Kohn-Sham DFT equations
- avoids explicit construction of Green functions with proper boundary conditions

However, it is important to note that solving of BLV self-consistent equations within the relativistic EDF theory is also numerically challenging. Instead of solving for one self-consistent system of equations in nuclear densities, we have to perform variations with both $\rho^{Nucl+Vap}$ and ρ^{Vap} , which leads to a coupled self-consistent system of equations.

5) *Second of all, they used DD-ME2, DD-PC1, DD-PCX. I think they made a good choice for the EDFs in this work. As I mentioned, they are quite consistent with the well known results*

for symmetric nuclear matter, and symmetry energy parameters. I think that the amount of works with three different models is enormously huge (for the calculation of driplines or mass tables) but three different models -which are the almost central point of nuclear matter properties won't be enough to generalized your results. Calculations from some models with different nuclear matter properties might make your work solid.

We thank the referee for his/her comments. To make our results more general we have extended our calculations to include four additional relativistic EDFs with different nuclear matter properties. In particular, we have selected the density-dependent point-coupling functionals (DD-PC) from Ref. [20], constrained to a specific value of the symmetry energy at saturation density (denoted by J): DD-PCJ30 ($J = 30$ MeV), DD-PCJ32 ($J = 32$ MeV), DD-PCJ34 ($J = 34$ MeV) and DD-PCJ36 ($J = 36$ MeV) functionals. We have chosen these particular functionals because the symmetry energy should have significant impact on the position of the drip-lines. We calculate the two-proton and two-neutron drip lines for five temperatures: $T = 0, 0.5, 1, 1.5$ and 2 MeV. Results of the calculations are shown in this reply in Fig. 1(a)–(e), and they are both commented in the revised main text of the manuscript (new Fig. 3 with the corresponding discussion on pages 10-12) and the revised Supplementary Information (Supplementary Note VII).

Indeed, we observe that calculations performed with additional DD-PCJ functionals predict systematic dependence of the neutron drip-line on the symmetry energy, and the same influence of the temperature on the position of drip-line as our previous calculations based on the DD-ME2, DD-PC1 and DD-PCX functionals. Although one could employ various different functionals, their number is limited by the available computational resources. However, we believe that functionals selected in the revised version of the manuscript provide robust conclusion about the influence of the temperature on the position of two-nucleon drip-line which is the main result of our work.

Supplementary Figure 1. The temperature evolution of two-nucleon drip lines at $T = 0, 0.5, 1, 1.5$ and 2 MeV. The calculations are performed with the density-dependent point-coupling functionals constrained to symmetry energy values $J = 30, 32, 34$ and 36 MeV.

6) *In the same matter, the study of different temperature is limited to $T=0.0\text{MeV}$, 0.5MeV , 1.0MeV , and 2.0MeV . More temperature grid points are necessary to generalized your results. To support your arguments such as, "At $T = 2.0 \text{ MeV}$, shell effects are completely washed out, nuclei are mostly spherical, and the neutron drip line is described by a simple straight line. " or "However, with increasing temperature at $T = 2 \text{ MeV}$ a dramatic change occurs, and most nuclei become spherical." I think you need more grid points between 1MeV and 2MeV .*

Based on the referee's suggestion we have increased the number of considered temperature mesh-points. In addition to previous set of $T = 0, 0.5, 1$ and 2 MeV , now we also consider $T = 0.8, 1.2, 1.5$ and 1.8 MeV , doubling the data-set of our calculations. All calculations are performed with DD-ME2, DD-PC1 and DD-PCX functionals. Results are shown in Supplementary Figure 10, together with tabulated locations of two-nucleon drip lines for all considered functionals and temperatures in Supplementary Information. Although the main conclusions of this work regarding the influence of temperature on the drip lines are better visualized at the original points of $T = 0, 0.5, 1$ and 2 MeV , the interested reader can investigate effects on a more finer temperature mesh in the Supplementary Information.

7) *I agree with your observation that the deformation vanishes as temperature increases. I'm also wondering if you can show the plots (beta_2- yaxis, Temperature-xaxis) to support your results.*

We show the temperature evolution of the isoscalar quadrupole deformation β_2^{IS} (defined in Eq.(S32) of the updated Supplementary Information) for selected even-even isotopes of neodymium here in Fig. 2 up to 3 MeV , with three relativistic EDFs. Guided by the referee's question, the corresponding discussion together with the figure is included in Supplementary Note IV on pages 20-22 of the updated Supplementary Information. For most nuclei, the shape phase-transition from the axially-deformed to a spherical shape occurs between $T = 1$ and $T = 3 \text{ MeV}$, depending on the underlying shell-structure of the nucleus.

8) *In the supplementary materials, your paper says "DD-PCX tends to predict the two-neutron dripline at a lower number of neutron N ." Why does it happen? Was it caused by the difference from the different pairing parameters?*

Supplementary Figure 2. Evolution of the isoscalar quadrupole deformation β_2^{IS} as a function of the temperature T for ^{150}Nd (a), ^{170}Nd (b), ^{180}Nd (c), and ^{190}Nd (d). Calculations are performed with DD-ME2 (solid blue), DD-PC1 (solid orange) and DD-PCX (solid green) relativistic EDFs.

In our work, the same type of pairing interaction (separable pairing) has been used in the calculations. However, it should be noted that different nuclear EDFs use different functional forms and parameter sets to describe nuclear properties. Additionally, the parameters of each relativistic EDF (including also pairing strength) are constrained using a distinct and limited set of experimental data, which can influence the predicted location of the drip lines. Some EDFs may also include different types of interactions, such as three-body forces or tensor forces, that are not included in other EDFs, which can lead to different predictions for the drip lines. Overall, the location of the nuclear drip lines is a complex and multifaceted problem that depends on many factors, and different nuclear EDFs can yield different predictions based on their specific assumptions and parameter sets.

Nonetheless, based on reply in point 5), where it has been shown that the location of the drip lines systematically varies with the symmetry energy associated to the EDF under consideration, we can also comment here on the differences between the drip line predictions from different EDFs. The DD-PCX functional has symmetry energy value $J = 31.3$ MeV

around saturation density, which is lower compared to both DD-PC1 and DD-ME2 which are constrained to $J = 33$ MeV and $J = 32.3$ MeV, respectively [21, 22]. Since a lower value of J leads to less neutron-rich nuclei, the two-neutron drip line is reached earlier for the DD-PCX, compared to the DD-ME2 and DD-PC1. It is interesting to note that differences in the location of the two-neutron drip line between the DD-ME2 and DD-PC1 are less pronounced, since J are more comparable. We explained this point in Supplementary Note VII.

9) *What kind of pairing models did you use in this work? If there is a specific pairing model for each EDF, why is there specific model? If not, why?*

In this work, for all considered functionals, we employ the same form of the so-called separable pairing interaction introduced in Refs. [23, 24], defined as

$$v^{pp}(1, 2) = -G\delta(\mathbf{R} - \mathbf{R}')P(\mathbf{r})P(\mathbf{r}'), \quad (\text{S5})$$

where $\mathbf{R} = 1/2(\mathbf{r}_1 + \mathbf{r}_2)$ and $\mathbf{r} = \mathbf{r}_1 - \mathbf{r}_2$ denote the center-of-mass and relative coordinate, respectively, while $P(\mathbf{r})$ has the form

$$P(\mathbf{r}) = \frac{1}{(4\pi a^2)^{3/2}} e^{-r^2/4a^2}. \quad (\text{S6})$$

The parameters G and a for the DD-ME2 and DD-PC1 interactions are taken from Ref. [23], while the parameters of the DD-PCX are from [25]. Based on the referee's question we have added the corresponding discussion at the end of Supplementary Note IC.

The main reason for using the separable pairing interaction lies in its numerical efficiency. By working in the basis of axially-deformed harmonic oscillator, the matrix element of v^{pp} can be written as a sum of the product of separable terms, which can be pre-calculated before the self-consistent solution procedure. In the original parameterization from Refs. [23, 24] the pairing strength G and the width a , were fitted to pairing gaps obtained with the pairing part of the Gogny D1S interaction [26]. Most implementations of relativistic EDFs have obtained successful results of nuclear properties across the nuclide chart by employing the pairing from the Gogny D1S interaction. However, its numerical calculation is significantly more computationally involving compared to the separable interaction, while the results are comparable.

10) *Maybe same question, "If N_{nucl} denotes the number of even-even nuclei between the*

Supplementary Figure 3. Temperature dependence of the number of bound even-even nuclei N_{nucl} as calculated with DD-PCJ30, DD-PCJ32, DD-PCJ34 and DD-PCJ36 relativistic EDFs.

drip lines, then it can be inferred that $N_{nucl}(DD-PC1) > N_{nucl}(DD-ME2) > N_{nucl}(DD-PCX)$, for all studied temperatures.” What is the fundamental reason for this?

The answer to this question is already covered in answers to points **5)** and **8)**. Since the DD-PCX interaction ($J = 31.3$ MeV) has lower symmetry energy value J compared to DD-ME2 ($J = 32.3$ MeV) and DD-PC1 ($J = 33$ MeV), by inspecting Fig. 3 in this reply we can infer that DD-PCX is going to predict less bound nuclei within the drip lines. The difference in the number of bound nuclei between DD-ME2 and DD-PC1 functionals is less compared to that of DD-PCX since the corresponding difference in J values is smaller than 1 MeV. Since we want to stress that the difference in the number of bound nuclei is larger for DD-PCX, we modify the inequality as $N_{nucl}(DD-PC1) \gtrsim N_{nucl}(DD-ME2) > N_{nucl}(DD-PCX)$.

11) *In your numerical work, the harmonic oscillator basis(HO) is used. But I’m wondering if it is adequate when there exist vapors. For example, HO is used for bound nuclei*

and if "HO" is used, most nuclei is morelike tightly bound. Thus to study halo nuclei, the transformed harmonic oscillator(THO) basis gives better results. How can you justify your results using HO basis?

We agree with the referee on the importance of verifying that the HO basis is adequate to treat the system consisting of nucleus and vapor. The THO basis, developed in order to remedy the wrong asymptotic behaviour of the HO wave functions (e^{-r^2} instead of e^{-r}), was mostly applied to the nonrelativistic functionals with publicly available HFBTHO solver [27–30]. On the other hand, applications to the relativistic functionals were rare and limited to spherical systems only [31]. In order to verify our HO solver supplemented with the BLV subtraction procedure, we have modified the coordinate-space solver based on the B-spline finite-element method (FEM) that was available to us, to solve the relativistic mean-field equations with the DD-ME2 interactions, assuming spherical symmetry (FT-RMFBSPL solver). Detailed description of the FT-RMFBSPL solver can be found in the updated version of the Supplementary Information, Supplementary Notes I and II. A comparison of the results obtained with the HO basis solver (FT-RMFHO) and the coordinate-space solver based on B-spline FEM (FT-RMFBSPL) is shown in Supplementary Figure 5 (Fig. 1 in the original version). We obtained excellent agreement between the FT-RMFBSPL solver and the FT-RMFHO solver with either $N_{osc} = 20$ and $N_{osc} = 60$ major oscillator shells used in the expansion of the Dirac spinors. Although the neutron RMS radii show slight discrepancies as the drip-line is approached, this does not influence the position of the drip-lines. We emphasise that the BLV subtraction procedure was implemented in all calculations presented in Supplementary Figure 5 and only in this case one can achieve the convergence of the calculation with respect to the dimension of the HO basis (FT-RMFHO solver) or the dimension of the box in the coordinate space (FT-RMFBSPL solver). The convergence properties of the BLV method have been verified with the coordinate-space solver in Refs. [32–34]. Furthermore, results were compared at $T = 2$ MeV temperature, where the pairing properties vanish and deformation effects are mostly washed out, in order to stress the contribution of vapor states. Finally, in this work we are not interested in studying halos, or observables for which the tail region of the density is crucial (e.g. radii for neutron rich nuclei). We are rather interested in the bulk nuclear properties (e.g. binding energies, deformations) for which the HO basis with BLV subtraction provides reliable

results.

As a further test, in Supplementary Figure 8 we show a comparison of the neutron emission lifetimes τ_n at $T = 2$ MeV and $T = 3$ MeV, with both the FT-RMFBSPL and FT-RMFHO solvers. Detailed description of the numerical calculation of the neutron emission lifetimes are described in Supplementary Note III of the revised Supplementary Information and additionally, in Supplementary table I, we have compared our results calculated with the axially-deformed HO solver (FT-DIRHBz) with the results presented in Ref. [35], which uses the axially-deformed coordinate-space solver with non-relativistic interactions. The results for both the neutron emission widths Γ as well as the neutron vapor density n_{gas} agree well [cf. Eq. (S29) in the revised Supp. Inf.].

Reviewer #3

1) *I find this a very interesting piece of work worth of being considered for publication. I would support publication as it is. The article is well written and to the point with several interesting physics messages about drippiness and the influence of temperature, with great consequences for the modeling of the synthesis of nuclei.*

We thank the referee for positive recommendation for publication of our work. We note that following comments by other reviewers we have revised and further improved our manuscript, with the modifications in the text highlighted in red color.

[1] E. Yüksel, **Temperature dependence of nuclear properties: A systematic study along the isotopic and isotonic chains of nuclei**, Nuclear Physics A 1014 (2021) 122238. doi:<https://doi.org/10.1016/j.nuclphysa.2021.122238>.

URL <https://www.sciencedirect.com/science/article/pii/S0375947421001032>

[2] F. Barranco, P. F. Bortignon, R. A. Broglia, G. Colò, P. Schuck, E. Vigezzi, X. Viñas, **Pairing matrix elements and pairing gaps with bare, effective, and induced interactions**, Phys. Rev. C 72 (2005) 054314. doi:[10.1103/PhysRevC.72.054314](https://doi.org/10.1103/PhysRevC.72.054314).

URL <https://link.aps.org/doi/10.1103/PhysRevC.72.054314>

[3] E. Litvinova, P. Ring, **Covariant theory of particle-vibrational coupling and its effect on the**

- single-particle spectrum, Phys. Rev. C 73 (2006) 044328. doi:10.1103/PhysRevC.73.044328.
URL <https://link.aps.org/doi/10.1103/PhysRevC.73.044328>
- [4] F. Barranco, R. A. Broglia, G. Gori, E. Vigezzi, P. F. Bortignon, J. Terasaki, Surface vibrations and the pairing interaction in nuclei, Phys. Rev. Lett. 83 (1999) 2147–2150. doi:10.1103/PhysRevLett.83.2147.
URL <https://link.aps.org/doi/10.1103/PhysRevLett.83.2147>
- [5] E. Litvinova, P. Schuck, Many-body correlations in nuclear superfluidity, Phys. Rev. C 102 (2020) 034310. doi:10.1103/PhysRevC.102.034310.
URL <https://link.aps.org/doi/10.1103/PhysRevC.102.034310>
- [6] H. Wibowo, E. Litvinova, Y. Zhang, P. Finelli, Temperature evolution of the nuclear shell structure and the dynamical nucleon effective mass, Phys. Rev. C 102 (2020) 054321. doi:10.1103/PhysRevC.102.054321.
URL <https://link.aps.org/doi/10.1103/PhysRevC.102.054321>
- [7] E. Litvinova, P. Schuck, Nuclear superfluidity at finite temperature, Phys. Rev. C 104 (2021) 044330. doi:10.1103/PhysRevC.104.044330.
URL <https://link.aps.org/doi/10.1103/PhysRevC.104.044330>
- [8] H. Wibowo, E. Litvinova, Nuclear shell structure in a finite-temperature relativistic framework, Phys. Rev. C 106 (2022) 044304. doi:10.1103/PhysRevC.106.044304.
URL <https://link.aps.org/doi/10.1103/PhysRevC.106.044304>
- [9] J. L. Egido, P. Ring, The decay of hot nuclei, Journal of Physics G: Nuclear and Particle Physics 19 (1) (1993) 1–54. doi:10.1088/0954-3899/19/1/002.
URL <https://doi.org/10.1088/0954-3899/19/1/002>
- [10] J. Egido, C. Dorso, J. Rasmussen, P. Ring, The nuclear deformation parameters at high excitation energies, Physics Letters B 178 (2) (1986) 139–144. doi:https://doi.org/10.1016/0370-2693(86)91484-X.
URL <https://www.sciencedirect.com/science/article/pii/037026938691484X>
- [11] L. Moretto, Pairing fluctuations in excited nuclei and the absence of a second order phase transition, Physics Letters B 40 (1) (1972) 1–4. doi:https://doi.org/10.1016/0370-2693(72)90265-1.
URL <https://www.sciencedirect.com/science/article/pii/0370269372902651>
- [12] A. L. Goodman, Statistical fluctuations in the $i_{13/2}$ model, Phys. Rev. C 29 (1984) 1887–1896.

- [doi:10.1103/PhysRevC.29.1887](https://doi.org/10.1103/PhysRevC.29.1887).
URL <https://link.aps.org/doi/10.1103/PhysRevC.29.1887>
- [13] V. Martin, J. L. Egido, Nuclear structure effects of the nuclei $^{152,154,156}\text{Dy}$ at high excitation energy and large angular momentum, Phys. Rev. C 51 (1995) 3084–3095. [doi:10.1103/PhysRevC.51.3084](https://doi.org/10.1103/PhysRevC.51.3084).
URL <https://link.aps.org/doi/10.1103/PhysRevC.51.3084>
- [14] V. Martin, J. L. Egido, L. M. Robledo, Thermal shape fluctuation effects in the description of hot nuclei, Phys. Rev. C 68 (2003) 034327. [doi:10.1103/PhysRevC.68.034327](https://doi.org/10.1103/PhysRevC.68.034327).
URL <https://link.aps.org/doi/10.1103/PhysRevC.68.034327>
- [15] S. Shlomo, G. Bertsch, Nuclear response in the continuum, Nuclear Physics A 243 (3) (1975) 507–518. [doi:https://doi.org/10.1016/0375-9474\(75\)90292-4](https://doi.org/10.1016/0375-9474(75)90292-4).
URL <https://www.sciencedirect.com/science/article/pii/0375947475902924>
- [16] X. Y. Qu, Y. Zhang, Canonical states in continuum skyrme hartree-fock-bogoliubov theory with green’s function method, Phys. Rev. C 99 (2019) 014314. [doi:10.1103/PhysRevC.99.014314](https://doi.org/10.1103/PhysRevC.99.014314).
URL <https://link.aps.org/doi/10.1103/PhysRevC.99.014314>
- [17] T. T. Sun, S. Q. Zhang, Y. Zhang, J. N. Hu, J. Meng, Green’s function method for single-particle resonant states in relativistic mean field theory, Phys. Rev. C 90 (2014) 054321. [doi:10.1103/PhysRevC.90.054321](https://doi.org/10.1103/PhysRevC.90.054321).
URL <https://link.aps.org/doi/10.1103/PhysRevC.90.054321>
- [18] A. T. Kruppa, M. Bender, W. Nazarewicz, P.-G. Reinhard, T. Vertse, S. Ówiok, Shell corrections of superheavy nuclei in self-consistent calculations, Phys. Rev. C 61 (2000) 034313. [doi:10.1103/PhysRevC.61.034313](https://doi.org/10.1103/PhysRevC.61.034313).
URL <https://link.aps.org/doi/10.1103/PhysRevC.61.034313>
- [19] T. Vertse, A. T. Kruppa, W. Nazarewicz, Shell corrections for finite-depth deformed potentials: Green’s function oscillator expansion method, Phys. Rev. C 61 (2000) 064317. [doi:10.1103/PhysRevC.61.064317](https://doi.org/10.1103/PhysRevC.61.064317).
URL <https://link.aps.org/doi/10.1103/PhysRevC.61.064317>
- [20] E. Yüksel, T. Oishi, N. Paar, Nuclear equation of state in the relativistic point-coupling model constrained by excitations in finite nuclei, Universe 7 (3) (2021). [doi:10.3390/universe7030071](https://doi.org/10.3390/universe7030071).

- URL <https://www.mdpi.com/2218-1997/7/3/71>
- [21] G. A. Lalazissis, T. Nikšić, D. Vretenar, P. Ring, *New relativistic mean-field interaction with density-dependent meson-nucleon couplings*, Phys. Rev. C 71 (2005) 024312. doi:10.1103/PhysRevC.71.024312.
URL <https://link.aps.org/doi/10.1103/PhysRevC.71.024312>
- [22] T. Nikšić, D. Vretenar, P. Ring, *Relativistic nuclear energy density functionals: Adjusting parameters to binding energies*, Phys. Rev. C 78 (2008) 034318. doi:10.1103/PhysRevC.78.034318.
URL <https://link.aps.org/doi/10.1103/PhysRevC.78.034318>
- [23] Y. Tian, Z.-y. Ma, P. Ring, *Axially deformed relativistic Hartree Bogoliubov theory with a separable pairing force*, Phys. Rev. C 80 (2009) 024313. doi:10.1103/PhysRevC.80.024313.
URL <https://link.aps.org/doi/10.1103/PhysRevC.80.024313>
- [24] Y. Tian, Z.-y. Ma, P. Ring, *Separable pairing force for relativistic quasiparticle random-phase approximation*, Phys. Rev. C 79 (2009) 064301. doi:10.1103/PhysRevC.79.064301.
URL <https://link.aps.org/doi/10.1103/PhysRevC.79.064301>
- [25] E. Yüksel, T. Marketin, N. Paar, *Optimizing the relativistic energy density functional with nuclear ground state and collective excitation properties*, Phys. Rev. C 99 (2019) 034318. doi:10.1103/PhysRevC.99.034318.
URL <https://link.aps.org/doi/10.1103/PhysRevC.99.034318>
- [26] D. Vretenar, A. Afanasjev, G. Lalazissis, P. Ring, *Relativistic hartree–bogoliubov theory: static and dynamic aspects of exotic nuclear structure*, Physics Reports 409 (3) (2005) 101–259. doi:<https://doi.org/10.1016/j.physrep.2004.10.001>.
URL <https://www.sciencedirect.com/science/article/pii/S0370157304004545>
- [27] M. Stoitsov, J. Dobaczewski, W. Nazarewicz, P. Ring, *Axially deformed solution of the skyrme–hartree–fock–bogolyubov equations using the transformed harmonic oscillator basis. the program hfbtho (v1.66p)*, Computer Physics Communications 167 (1) (2005) 43–63. doi:<https://doi.org/10.1016/j.cpc.2005.01.001>.
URL <https://www.sciencedirect.com/science/article/pii/S0010465505000305>
- [28] M. Stoitsov, N. Schunck, M. Kortelainen, N. Michel, H. Nam, E. Olsen, J. Sarich, S. Wild, *Axially deformed solution of the skyrme–hartree–fock–bogoliubov equations using the transformed harmonic oscillator basis (ii) hfbtho v2.00d: A new version of the program*, Computer

- Physics Communications 184 (6) (2013) 1592–1604. doi:<https://doi.org/10.1016/j.cpc.2013.01.013>.
 URL <https://www.sciencedirect.com/science/article/pii/S0010465513000301>
- [29] R. N. Perez, N. Schunck, R.-D. Lasserri, C. Zhang, J. Sarich, *Axially deformed solution of the skyrme–hartree–fock–bogolyubov equations using the transformed harmonic oscillator basis (iii) hfbtho (v3.00): A new version of the program*, Computer Physics Communications 220 (2017) 363–375. doi:<https://doi.org/10.1016/j.cpc.2017.06.022>.
 URL <https://www.sciencedirect.com/science/article/pii/S0010465517302047>
- [30] P. Marević, N. Schunck, E. Ney, R. Navarro Pérez, M. Verriere, J. O’Neal, *Axially-deformed solution of the skyrme-hartree-fock-bogoliubov equations using the transformed harmonic oscillator basis (iv) hfbtho (v4.0): A new version of the program*, Computer Physics Communications 276 (2022) 108367. doi:<https://doi.org/10.1016/j.cpc.2022.108367>.
 URL <https://www.sciencedirect.com/science/article/pii/S0010465522000868>
- [31] M. Stoitsov, P. Ring, D. Vretenar, G. A. Lalazissis, *Solution of relativistic hartree-bogoliubov equations in configurational representation: Spherical neutron halo nuclei*, Phys. Rev. C 58 (1998) 2086–2091. doi:[10.1103/PhysRevC.58.2086](https://doi.org/10.1103/PhysRevC.58.2086).
 URL <https://link.aps.org/doi/10.1103/PhysRevC.58.2086>
- [32] E. Suraud, *Semi-classical calculations of hot nuclei*, Nuclear Physics A 462 (1) (1987) 109–149. doi:[https://doi.org/10.1016/0375-9474\(87\)90382-4](https://doi.org/10.1016/0375-9474(87)90382-4).
 URL <https://www.sciencedirect.com/science/article/pii/0375947487903824>
- [33] P. Bonche, S. Levit, D. Vautherin, *Properties of highly excited nuclei*, Nuclear Physics A 427 (2) (1984) 278–296. doi:[https://doi.org/10.1016/0375-9474\(84\)90086-1](https://doi.org/10.1016/0375-9474(84)90086-1).
 URL <https://www.sciencedirect.com/science/article/pii/0375947484900861>
- [34] P. Bonche, S. Levit, D. Vautherin, *Statistical properties and stability of hot nuclei*, Nuclear Physics A 436 (2) (1985) 265–293. doi:[https://doi.org/10.1016/0375-9474\(85\)90199-X](https://doi.org/10.1016/0375-9474(85)90199-X).
 URL <https://www.sciencedirect.com/science/article/pii/037594748590199X>
- [35] Y. Zhu, J. C. Pei, *Microscopic description of neutron emission rates in compound nuclei*, Phys. Rev. C 90 (2014) 054316. doi:[10.1103/PhysRevC.90.054316](https://doi.org/10.1103/PhysRevC.90.054316).
 URL <https://link.aps.org/doi/10.1103/PhysRevC.90.054316>

REVIEWER COMMENTS

Reviewer #1 (Remarks to the Author):

The authors have carefully answered the questions put by the referees, modifying and extending the manuscript accordingly, also carrying out new calculations.

In my opinion, the paper has been considerably improved and deserves publication.

I have a few remarks that should be easily addressed.

a) The authors write in the Abstract that “the total number of bound nuclei is larger at high temperatures compared to cold nuclei”. This statement is supported by the results shown in Fig. 1 for the $Z=60$ isotopic chain. I am not very convinced by this statement, in view of the final Tables reported in the Supplementary material, where one can notice that there are isotopic chains for which the number of bound nuclei decreases (see for example $Z=76$) or stays constant also for the functionals studied in the main text. In Table II of the Supplementary material one can see that the total increase in the number of bound nuclei is equal to a few per cent. On the other hand the new Fig. 13 in the Supplementary file shows that number of bound nuclei slightly decreases for $T=1$ MeV and then at $T=2$ MeV is more or less the same as for $T=0$ for all the four DDPC functionals shown there.

Probably it would be more fair to state that “The total number of bound nuclei remains essentially constant as a function of the temperature”. In any case, it is clear from Fig. 13 and from the discussion about symmetry energy in the main text, that the number of bound nuclei is much more sensitive to the value of J than to the temperature. I would suggest to the authors to show Fig. 13 in the main text, including also

the results shown in Table II, in order for the reader to assess in a compact way the influence of the temperature and of the symmetry energy on the number of bound nuclei.

I would also ask the authors to detail briefly what are the “simple arguments” referred to on p.9 that would make one expect a smaller number of bound nuclei at larger temperature.

b) It is not clear to me what is meant by ‘essential treatment’ in the Abstract

c) The sentence “Extreme astrophysical events ... influence nuclear properties” on p.3 is essentially repeated three lines below (“Therefore ... finite temperatures”).

d) The scale of the two panels of Fig. 2 is not the same, making it difficult to compare them.

I would suggest to show the $T=1$ and $T=2$ MeV results in the same panel, showing simultaneously the effect

of the temperature and of the subtraction procedure.

e) It is not clear to me why the author conclude at the end of p.16 that “the inclusion of both beyond-mean field approaches and thermal fluctuations in the calculations can lead to slight changes in the predictions of the drip lines at zero and finite temperature”, while a few lines before they had written that “the impact of beyond mean field approaches can be important for the calculations of drip lines at zero temperatures”.

Concerning this point, I would suggest to insert the paper by A. Idini et al., PRC 85 (2012) 014331 in the references.

f) It is not clear what is meant by "initialized" in the last lines of p.17.

g) Does "boson and fermion states" on p.19 stand for 'neutrons and protons'?

Reviewer #2 (Remarks to the Author):

I'm pleased to see the reply note of the authors to my questions and requests.

They answered to all my questions in details.

They also did a lot of numerical works to check all my requests.

It is well deserved to be published in the nature communications.

One thing that I don't agree with their results is that

"Since a lower value of J leads to less neutron-rich nuclei, the two-neutron drip line is reached earlier for the DD-PCX, compared to the DD-ME2 and DD-PC1"

-> This might be wrong sentence.

The magnitude of symmetry energy implies how tightly neutrons and protons are bounded. That means, the large symmetry energy(J) means nuclei prefer to have equal number of neutrons and protons.

It can be easily checked with the liquid drop model approaches.

I guess the pairing formalism in different EDFs caused such differences in neutron driplines instead of the symmetry energy differences.

Other than that, I think it is good enough to be published in the journal.

SECOND REPLY TO THE REVIEWERS' COMMENTS ON MANUSCRIPT

NCOMMS-22-49107

We thank the reviewers for reading our revised version of the paper, positive recommendations, and additional suggestions. By considering the reviewers' comments, we have further revised the manuscript as follows.

Reviewer #1

1) *The authors write in the Abstract that “the total number of bound nuclei is larger at high temperatures compared to cold nuclei”. This statement is supported by the results shown in Fig. 1 for the Z=60 isotopic chain. I am not very convinced by this statement, in view of the final Tables reported in the Supplementary material, where one can notice that there are isotopic chains for which the number of bound nuclei decreases (see for example Z=76) or stays constant also for the functionals studied in the main text. In Table II of the Supplementary material one can see that the total increase is in the number of bound nuclei is equal to a few per cent. On the other hand the new Fig. 13 in the Supplementary file shows that number of bound nuclei slightly decreases for T=1 MeV and then at T=2 MeV is more or less the same as for T=0 for all the four DDPC functionals shown there. Probably it would be more fair to state that “The total number of bound nuclei remains essentially constant as a function of the temperature”. In any case, it is clear from Fig. 13 and from the discussion about symmetry energy in the main text, that the number of bound nuclei is much more sensitive to the value of J than to the temperature. I would suggest to the authors to show Fig. 13 in the main text, including also the results shown in Table II, in order for the reader to assess in a compact way the influence of the temperature and of the symmetry energy on the number of bound nuclei. I would also ask the authors to detail briefly what are the “simple arguments” referred to on p.9 that would make one expect a smaller number of bound nuclei at larger temperature.*

Following reviewer's suggestion, we have included Fig. 1 shown in this reply to the main manuscript, together with the complementary discussion added on Page 13:

”In Figure 4(a), we demonstrate the variation of the number of bound nuclei N_{nucl} with temperature for seven relativistic EDFs considered in this work. For a set of DD-PCJ functionals constrained to a specific symmetry energy J a clear hierarchy is observed. Functionals with lower J tend to predict a smaller number of bound nuclei for all considered temperatures. Although DD-ME2 ($J = 32.3$ MeV), DD-PC1 ($J = 33$ MeV), and DD-PCX ($J = 31.1$ MeV) have distinct formulations and optimisation procedures employed to constrain their parameters, we also observe a comparable dependence on the symmetry energy values. However, a more detailed comparison requires investigating the full density-dependence of the symmetry energy S_2 . To provide a more comprehensible representation of the variations in the number of bound nuclei as temperature increases, we present in Figure 4(b) the relative change of N_{nucl} with respect to zero temperature. At low temperatures, the interplay between the properties of nuclear effective interaction, pairing, and temperature effects determines the nuclear binding and number of bound nuclei, exhibiting a nearly constant or slightly decreasing trend. As the temperature is raised to $T=1$ MeV, the number of bound nuclei undergoes variation, either with a slight increase or decrease, depending upon the specific functional employed in the calculation. At higher temperatures, the pairing correlations disappear, shell effects vanish, and the number of bound nuclei starts to increase with increasing temperature for all functionals considered in this work.”

Due to these changes, we also slightly modified the abstract and added the following sentences:

”At temperatures $T \lesssim 12$ billion kelvins, the interplay between the properties of nuclear effective interaction, pairing, and temperature effects determines the nuclear binding. At higher temperatures, we find a surprising result that the total number of bound nuclei increases with temperature due to thermal shell quenching.”

We also modified the last sentence in the Introduction on Page 5:

”Surprisingly, our analysis reveals that at higher temperatures the nuclear landscape is expanding, i.e., at $T \gtrsim 1$ MeV the total number of bound nuclei start to increase because of the thermal quenching of the shell effects.”

Considering the reviewer’s comment related to ”simple arguments”, we also modified the following sentence starting on line 182:

”Since nuclei become less bound with increasing temperature [1–3], one could expect that at finite temperatures the overall number of bound nuclei within the nuclide chart should become smaller.”

FIG. 1. (a) Temperature evolution of the total number of bound nuclei N_{nucl} for seven relativistic EDFs. (b) Relative change of N_{nucl} with temperature.

2) *It is not clear to me what is meant by ‘essential treatment’ in the Abstract*

To avoid confusion, we removed word ‘essential’ from the abstract. We modified the sentence on line 15 as follows:

”...including treatment of thermal scattering of nucleons in the continuum.”

3) *The sentence “Extreme astrophysical events ... influence nuclear properties” on p.3 is essentially repeated three lines below (“Therefore ... finite temperatures”).*

The sentence “Therefore, a comprehensive understanding of extreme astrophysical phenomena also requires precise knowledge of nuclear properties at finite temperatures.” is replaced by (line 44):

”Complete understanding of all these phenomena requires knowledge of nuclear properties at finite temperature.”

4) *The scale of the two panels of Fig. 2 is not the same, making it difficult to compare them. I would suggest to show the $T=1$ and $T=2$ MeV results in the same panel, showing simultaneously the effect of the temperature and of the subtraction procedure.*

We have modified Fig. 2 in order to have the results shown on one panel. The resulting figure is Fig. 2 in this reply.

5) *It is not clear to me why the author conclude at the end of p.16 that “the inclusion of both beyond-mean field approaches and thermal fluctuations in the calculations can lead to slight changes in the predictions of the drip lines at zero and finite temperature”, while a few lines before they had written that “the impact of beyond mean field approaches can be important for the calculations of drip lines at zero temperatures”. Concerning this point, I would suggest to insert the paper by A. Idini et al., PRC 85 (2012) 014331 in the references.*

As we explained in the previous version of the main manuscript, we expect more pronounced effects of beyond mean-field effects on the drip line at lower temperatures due to enhanced pairing properties of nuclei. In fact, the induced pairing interaction due to phonon couplings is comparable to the bare pairing interaction, as mentioned in the work of A. Idini et al. in Ref. [4] as well as references cited in the main paper. At higher temperatures, when pairing collapses, we expect beyond mean-field effects to lead to more modest changes. This is supported by work in Ref. [5] where it can be seen that single-particle level fragmentation is more pronounced for deeply-bound states compared to those near the Fermi level at $T > 1$ MeV. However, the quantitative assessment of these effects on our results for the drip lines

FIG. 2. The new Fig. 2 of the main manuscript showing the impact of continuum subtraction at $T = 1$ and 2 MeV.

would require thorough study that goes beyond our theory framework. Thus, to avoid any confusion in the text, we removed the last sentences starting with:

”Thus, the inclusion of both beyond-mean field approaches... ”

As suggested, we also added the following paper to the References: A. Idini et al., PRC 85 (2012) 014331.

6) *It is not clear what is meant by "initialized" in the last lines of p.17.*

Before starting the self-consistent procedure to solve the FT-RHB equations, one has to initialize both the mean-field h and the pairing field Δ . Concerning the mean-field h , the Woods-Saxon potential form is assumed for the vector and scalar fields, while the Coulomb field assumes a homogeneous distribution of charge. The pairing field Δ is initialized by a

constant pairing gap. The peculiarity of the BLV procedure is that we have two systems to initialize. First, the Nucleus+Vapor(N+V) system is initialized as already described (usual prescription assumed in these models), while the vapor(V) system is initialized only with the Coulomb field (having the same form as that for the N+V system). After the initialization step, we can start the self-consistent solution procedure for the two coupled systems.

To clarify this in the text, we modified sentences starting at line 351 (page 19):

”The resulting coupled FT-RHB equations are solved through self-consistent iterations by utilizing the modified Broyden mixing [6]. Before starting the self-consistent procedure the mean- and pairing-field should be initialized. The N+V system follows the usual prescription as in Ref. [6], where the nuclear mean-field assumes a combination of the Woods-Saxon and the Coulomb potential. On the other hand, the V system is only initialized by the Coulomb potential.”

7) *Does “boson and fermion states” on p.19 stand for ‘neutrons and protons’?*

Fermion states refer to Dirac spinors (nucleons) while boson states correspond to wave-functions of σ , ω and ρ mesons. We note that only the DD-ME2 interaction contains meson fields and therefore number of oscillator shells for the bosons is a parameter. On the other hand, the DD-PC functionals are characterized only by the number of fermion shells since the meson propagators are replaced with contact interaction vertices.

To clarify reviewer’s point we have modified the first sentence starting on line 382 (page 20) from:

”Unless stated otherwise, calculations are performed with 20 oscillator shells for both fermion and boson states, which yields satisfying convergence properties for our finite temperature study.”

to:

”The FT-RHB equations are solved by expanding the wavefunctions and meson fields in the basis of axially-deformed harmonic oscillator characterized by N_{osc} shells. Both fermion (nucleon Dirac spinors) and boson (minimal set of σ , ω and ρ mesons appearing only in the DD-ME2 EDF) states are expanded in $N_{osc} = 20$, which yields satisfying convergence properties for our finite temperature study.”

Reviewer #2

1) *One thing that I don't agree with their results is that "Since a lower value of J leads to less neutron-rich nuclei, the two-neutron drip line is reached earlier for the DD-PCX, compared to the DD-ME2 and DD-PC1". This might be wrong sentence. The magnitude of symmetry energy implies how tightly neutrons and protons are bounded. That means, the large symmetry energy(J) means nuclei prefer to have equal number of neutrons and protons. It can be easily checked with the liquid drop model approaches. I guess the pairing formalism in different EDFs caused such differences in neutron driplines instead of the symmetry energy differences.*

We thank to the reviewer for positive feedback on our work. We note that although the liquid drop model (LDM) offers a qualitative description of the nuclear binding energy, one cannot always compare the LDM parameters directly to the nuclear matter properties calculated within the EDF framework. The LDM considers the atomic nucleus as an incompressible liquid drop of uniform density. On the other hand, the EDF based calculation provides the symmetry energy as a function of density.

Motivated by the reviewer's remark, we made additional investigation to further understand the influence of symmetry energy on the drip line.

First, in Fig. 3(a) we show the number of bound nuclei N_{nucl} at zero temperature as a function of the symmetry energy J at saturation density. A clear increasing trend is observed for all considered functionals. We can immediately discern the influence of pairing correlations on the observed trend by raising the temperature above the pairing collapse temperature (which is around 1 MeV). Therefore, in Fig. 3(b) we show the dependence of N_{nucl} on J at $T = 2$ MeV. Indeed, a similar trend with J is observed.

FIG. 3. The number of bound nuclei as function of the symmetry energy at saturation density J for a specific relativistic EDFs at $T = 0$ (a) and $T = 2$ MeV (b).

However, the symmetry energy at saturation density J represents only a zeroth-order term and does not portray a full picture. To better understand previous results, in Fig. 4 we show the density dependence of the symmetry energy $S_2(\rho)$ of the symmetric nuclear matter for DD-PCJ [7] family of functionals in addition to DD-PC1 and DD-PCX. One can see that there exists a point $\rho_{insc} \sim 0.11 \text{ fm}^{-3}$ where symmetry energies of DD-PCJ functionals intersect. This is especially interesting since for $\rho > \rho_{insc}$ the symmetry energy S_2 shows an increasing behaviour with J while for those below $\rho < \rho_{insc}$ we have a decreasing trend.

First, we can select a density that satisfies $\rho_{insc} < \rho = 0.12 \text{ fm}^{-3}$, which is also known as the average nuclear density, and show dependence of N_{nucl} on $S_2(\rho = 0.12 \text{ fm}^{-3})$ in Fig. 5(a). For the DDPCJ family of functionals constrained to a specific J , a clear increasing trend of the number of bound nuclei with symmetry energy is shown. On the other hand, if we select $\rho_{insc} > \rho = 0.09 \text{ fm}^{-3}$, we obtain a sharply decreasing trend in a number of bound

FIG. 4. Density dependence of the symmetry energy $S_2(\rho)$ of the symmetric nuclear matter for DD-PC family of relativistic EDFs. Inserted figure shows enlarged regions where symmetry energies for different EDFs intersect. Calculations are performed at zero temperature.

nuclei with symmetry energy as shown in Fig. 5(b). Therefore, the question of increasing or decreasing trend of number of bound nuclei with symmetry energy depends on what is considered as the average density.

In our study, we specifically examine the symmetry energy and its slope around the saturation densities of $\rho_0 \approx 0.16 \text{ fm}^{-3}$. We find that an increase in the symmetry energy leads to a higher total number of bound nuclei. A previous study in Ref. [8] yields similar results to ours, as the authors also restrict the average density within the range of 0.14-0.17 fm^{-3} . In Ref. [9], the authors investigate the neutron drip line and its relationship with the symmetry energy using both relativistic and non-relativistic functionals. They focus on a region of the symmetry energy $S_2(\rho = \rho_{sc})$ at sub-saturation cross density ρ_{sc} , thus discovering a negative correlation between the number of bound nuclei and the symmetry energy.

Based on the present discussion we have revised our conclusions presented on page 29 of the Supplementary material where we claim:

FIG. 5. Number of bound nuclei as function of the symmetry energy S_2 at $\rho = 0.12 \text{ fm}^{-3}$ (a) and $\rho = 0.09 \text{ fm}^{-3}$ (b).

”Namely, since the DD-PC1 functional has a larger symmetry energy ($J = 33 \text{ MeV}$) compared to the DD-ME2 ($J = 32.3 \text{ MeV}$), and the DD-PCX ($J = 31.3 \text{ MeV}$), the two-neutron drip line as calculated with the DD-PC1 is going to be more neutron-rich.”

In the revised version of the Supplementary information on page 29 we state:

”Considering DD-PC1 ($J = 33 \text{ MeV}$), DD-ME2 ($J = 32.3 \text{ MeV}$), and DD-PCX ($J = 31.3 \text{ MeV}$) functionals, we find that a similar hierarchy as in DD-PCJ family with symmetry energy at saturation density J is found. However, DD-PCJ functionals are optimized using the same set of nuclei and nuclear matter properties with only J varied. On the other hand, DD-PC1, DD-PCX and DD-ME2 considerably differ in their optimization protocols and employed nuclear properties. Therefore, although our results show that those functionals follow similar trends as DD-PCJ set, detailed assessment of the correlation of N_{nucl} with J

requires investigation of the density dependence of symmetry energy $S_2(\rho)$.”

Furthermore, in the main manuscript we include the following discussion on page 13:

”Although DD-ME2 ($J = 32.3$ MeV), DD-PC1 ($J = 33$ MeV), and DD-PCX ($J = 31.1$ MeV) have distinct formulations and optimisation procedures employed to constrain their parameters, we also observe a comparable dependence on the symmetry energy values. However, a more detailed comparison requires investigating the full density-dependence of the symmetry energy S_2 .”

-
- [1] Y. F. Niu, Z. M. Niu, N. Paar, D. Vretenar, G. H. Wang, J. S. Bai, J. Meng, Pairing transitions in finite-temperature relativistic hartree-bogoliubov theory, Phys. Rev. C 88 (2013) 034308. doi:[10.1103/PhysRevC.88.034308](https://doi.org/10.1103/PhysRevC.88.034308).
URL <https://link.aps.org/doi/10.1103/PhysRevC.88.034308>
- [2] A. L. Goodman, Finite-temperature HFB theory, Nuclear Physics A 352 (1) (1981) 30–44. doi:[https://doi.org/10.1016/0375-9474\(81\)90557-1](https://doi.org/10.1016/0375-9474(81)90557-1).
URL <https://www.sciencedirect.com/science/article/pii/0375947481905571>
- [3] E. Yüksel, Temperature dependence of nuclear properties: A systematic study along the isotopic and isotonic chains of nuclei, Nuclear Physics A 1014 (2021) 122238. doi:<https://doi.org/10.1016/j.nuclphysa.2021.122238>.
URL <https://www.sciencedirect.com/science/article/pii/S0375947421001032>
- [4] A. Idini, F. Barranco, E. Vigezzi, Quasiparticle renormalization and pairing correlations in spherical superfluid nuclei, Phys. Rev. C 85 (2012) 014331. doi:[10.1103/PhysRevC.85.014331](https://doi.org/10.1103/PhysRevC.85.014331).
URL <https://link.aps.org/doi/10.1103/PhysRevC.85.014331>
- [5] H. Wibowo, E. Litvinova, Nuclear shell structure in a finite-temperature relativistic framework, Phys. Rev. C 106 (2022) 044304. doi:[10.1103/PhysRevC.106.044304](https://doi.org/10.1103/PhysRevC.106.044304).
URL <https://link.aps.org/doi/10.1103/PhysRevC.106.044304>
- [6] T. Nikšić, N. Paar, D. Vretenar, P. Ring, DIRHB–A relativistic self-consistent mean-field framework for atomic nuclei, Computer Physics Communications 185 (6) (2014) 1808–1821.

doi:<https://doi.org/10.1016/j.cpc.2014.02.027>.

URL <https://www.sciencedirect.com/science/article/pii/S0010465514000836>

- [7] E. Yüksel, T. Oishi, N. Paar, **Nuclear equation of state in the relativistic point-coupling model constrained by excitations in finite nuclei**, *Universe* 7 (3) (2021). doi:10.3390/universe7030071.

URL <https://www.mdpi.com/2218-1997/7/3/71>

- [8] K. Oyamatsu, K. Iida, H. Koura, **Neutron drip line and the equation of state of nuclear matter**, *Phys. Rev. C* 82 (2010) 027301. doi:10.1103/PhysRevC.82.027301.

URL <https://link.aps.org/doi/10.1103/PhysRevC.82.027301>

- [9] R. Wang, L.-W. Chen, **Positioning the neutron drip line and the r-process paths in the nuclear landscape**, *Phys. Rev. C* 92 (2015) 031303. doi:10.1103/PhysRevC.92.031303.

URL <https://link.aps.org/doi/10.1103/PhysRevC.92.031303>

REVIEWERS' COMMENTS

Reviewer #1 (Remarks to the Author):

The authors have answered my remarks satisfactorily, and in my opinion the manuscript can now be published.

Reviewer #2 (Remarks to the Author):

Thank you for the additional works to my question.

I'm fully satisfied with authors' present work.

THIRD REPLY TO THE REVIEWERS' COMMENTS ON MANUSCRIPT

NCOMMS-22-49107

We thank the reviewers for reading our final revision and positive recommendations for publication of our paper.

Reviewer #1 (Remarks to the Author):

The authors have answered my remarks satisfactorily, and in my opinion the manuscript can now be published.

Reviewer #2 (Remarks to the Author):

Thank you for the additional works to my question. I'm fully stastified with authors' present work.

Since there are no additional remarks by the reviewers, no further modifications in the manuscript are done, except those related to text formatting.